# Checkpoint kinase 1 is essential for normal B cell development and lymphomagenesis

Fabian Schuler[1], Johannes G. Weiss[1], Silke E. Lindner[1], Michael Lohmüller[1], Sebastian Herzog[1], Simon F. Spiegl [2], Philipp Menke [2], Stephan Geley[2], Verena Labi[1] & Andreas Villunger[1,3]

Checkpoint kinase 1 (CHK1) is critical for intrinsic cell cycle control and coordination of cell cycle progression in response to DNA damage. Despite its essential function, CHK1 has been identified as a target to kill cancer cells and studies using *Chk1* haploinsufficient mice initially suggested a role as tumor suppressor. Here, we report on the key role of CHK1 in normal B-cell development, lymphomagenesis and cell survival. Chemical CHK1 inhibition induces BCL2-regulated apoptosis in primary as well as malignant B-cells and CHK1 expression levels control the timing of lymphomagenesis in mice. Moreover, total ablation of *Chk1* in B-cells arrests their development at the pro-B cell stage, a block that, surprisingly, cannot be overcome by inhibition of mitochondrial apoptosis, as cell cycle arrest is initiated as an alternative fate to limit the spread of damaged DNA. Our findings define CHK1 as essential in B-cell development and potent target to treat blood cancer.

[1] Division of Developmental Immunology, Biocenter, Medical University of Innsbruck, Innrain 80, A-6020 Innsbruck, Austria. [2] Division of Molecular Pathophysiology, Biocenter, Medical University of Innsbruck, Innrain 80, A-6020 Innsbruck, Austria. [3] Tyrolean Cancer Research Institute, Innrain 66, A-6020 Innsbruck, Austria. Correspondence and requests for materials should be addressed to A.V. (email: andreas.villunger@i-med.ac.at)

Checkpoint kinases 1 (CHK1) and/or related CHK2 become activated as integral components of the DNA damage-response (DDR) machinery, able to resolve different types of DNA lesions[1]. CHK1 reacts to single-strand (ss) DNA breaks generated in response to genotoxic stress or replication errors during S-phase or at stalled replication forks[2]. ssDNA lesions become coated by Replication Protein A (RPA) and are recognized by Ataxia telangiectasia and Rad3 related (ATR) kinase that in turn phosphorylates CHK1 on $Ser^{317}$ and $Ser^{345}$ resulting in its activation[3]. Upon activation, CHK1 targets Cdc25A for degradation which dampens the activity of cyclin dependent kinase 2 (CDK2)/cyclin complexes slowing down DNA synthesis[4]. Upon DNA damage in G2, CHK1 blocks entry into mitosis by phosphorylation-dependent activation of Wee1 kinase, as well as by inhibition of Cdc25C phosphatase, both regulating inhibitory $Tyr^{15}$ phosphorylation on CDK1 and its subcellular localization[5–7]. Of note, in response to dsDNA breaks, the ATM/CHK2 signaling axis is key to solve the problem but CHK1 is also engaged as a result of resection of DNA double-strands at lesion sites, generating ssDNA intermediates[8]. Subsequently, CHK1 phosphorylates Rad51, a protein of the DDR, which promotes homologous recombination-mediated DNA repair[9].

During normal cell cycle progression CHK1 levels increase in an E2F-dependent manner during G1/S transition where it controls a number of events, including origin firing, elongation and replication-fork stability allowing for faithful DNA duplication[10–13]. In the absence of CHK1, cells enter mitosis prematurely in the presence of incompletely replicated DNA, leading to cell death by a so far ill-defined mechanism, often referred to as mitotic catastrophe[14,15]. CHK1 is inhibited at the normal G2/M transition by Polo-like kinase (PLK)1-mediated phosphorylation and degradation of claspin[16,17], a key regulator of CHK1 activity, as well as by CDK1/CyclinB-dependent inhibitory phosphorylation events in late G2, triggering nuclear export of CHK1 and therefore limiting substrate accessibility[18]. Active $Ser^{345}$ CHK1 has been detected in M-phase but its function in normal mitosis is less clear[2,19]. Yet, a number of substrates, including members of the spindle assembly check point machinery and several mitotic kinases, such as Aurora A/B and PLK1, have been described[20–22]. In summary, CHK1 has several important physiological functions in regulating DNA replication and repair, normal cell cycle progression and cell survival.

Consistent with these functions, a tumor suppressive role of CHK1 has been reported, mainly based on early mouse studies. While loss of *Chk1* was shown to be lethal, associated with impaired G2/M arrest and increased cell death in preimplantation embryos[14,15], transformation of mammary epithelial cells lacking one allele of *Chk1* using a MMTV-driven *Wnt1* transgene was demonstrated to be accelerated[23]. Furthermore, *Chk1* heterozygosity was shown to synergize with *p53* haplo-insufficiency in promoting breast cancer[24] and significantly accelerated the onset and tumor incidence in $Chk2^{-/-}$ mice that most frequently developed lymphoma[25]. Consistently, CHK1 mRNA and protein expression were reported to be reduced in aggressive variants of different human lymphoid malignancies[26]. However, *CHK1* deletion or homozygous loss-of-function mutations were not found in human cancer so far[26–28]. Notably, a number of solid cancer entities actually showed increased CHK1 expression, consistent with the idea that replication stress-associated DNA damage is a particular threat to cancer cells experiencing high oncogenic load[2,10,29]. In fact, mRNA levels are frequently elevated with highest levels found in fast proliferating lymphoma and leukemia cells. Of note, Burkitt lymphoma, a MYC-driven malignancy of germinal center B cells, shows highest mRNA levels across all cancers in the TGCA database, suggesting that high CHK1 activity might be needed to balance replication stress caused by deregulated MYC, or other oncogenic events that drive extensive proliferation[30]. In line with this hypothesis, *Eµ-MYC*-driven murine lymphomas are highly responsive to CHK1 inhibitors in transplant models and several human blood cancer cell lines, including Burkitt and Diffuse large B cell lymphomas (DLBCL) respond well to CHK1 inhibition alone or when combined with antimetabolites or ATR inhibitors[31–34].

Together, this suggests that CHK1 may exert tumor suppressive as well as oncogenic effects, depending on cellular context, transformation state or in response to different oncogenic cues. This may render some cancers addicted to CHK1 while reduced activity might facilitate transformation. For example, hematopoietic cancers that usually have a high proliferative index depend on a highly competent replication stress response (RSR) pathway[35]. Consistently, in a recent report it was shown that cytogenetically normal AML patients with high abundance of CHK1 expression have a poor prognosis and higher relapse rates. This was assigned to increased S-phase proficiency upon Cytarabine treatment that usually interferes with normal DNA synthesis and stalls replication forks, engaging CHK1 for stabilization. Co-treatment with the phase II clinical trial CHK1 inhibitor, SCH900776, dramatically increased the effect of Cytarabine treatment on AML[30]. A second study showed impressive efficacy of an ATR inhibitor, targeting the same vulnerability, as single agent in MLL-rearranged AML[36].

Together these studies highlight the potential of CHK1 inhibition to treat blood cancer in humans but it remains unclear if CHK1 function is needed mainly for transformation, tumor maintenance, or both. Furthermore, it is still uncertain under which conditions CHK1 may be tumor suppressive, or what would be the impact of CHK1 inhibition on normal hematopoiesis. Using chemical inhibition as well as genetic ablation of CHK1 in normal and malignant B cells, we demonstrate for the first time its key-role in lymphocyte transformation and tumor cell survival, as well as its essential contribution to normal B cell development. Together, our findings highlight its remarkable drug-target potential as well as putative side effects that may limit efficacy in clinical use.

## Results

### CHK1 inhibition kills Burkitt lymphoma and pre-B ALL cells.
CHK1 inhibitors have shown promising results in killing transplanted MYC-driven mouse B cell lymphomas in mice and Burkitt lymphoma cell lines in vitro. However, the mode of cell death induced was not clarified further in these studies[31,33]. Hence, we chose to interrogate a number of Burkitt lymphoma cell lines, assessed their CHK1 expression and activity levels (Fig. 1a) and treated them with the selective CHK1 inhibitors (CHKi), PF-477736 or CHIR-124 to assess their susceptibility to and mode of cell death. Cell lines were analyzed after 24 h (not shown) or 48 h, determining apoptosis and cell cycle status by Annexin-V binding or DNA-content analysis, respectively. All cell lines tested responded with increased cell death in a time (not shown) and dose-dependent manner with CHIR-124, being more potent than PF-477736 in its killing potential (Fig. 1b, Supplementary Fig. 1A). Of note, cell death was strongly reduced in the presence of the pan-caspase inhibitor, Q-VD-OPh (QVD), regardless of the type of CHK1 inhibitor used, indicative for apoptosis as the major route of cell death (Fig. 1c, Supplementary Fig. 1C). As most anticancer drugs trigger intrinsic apoptosis, we monitored the consequences of BCL2 overexpression in 5/10 cell lines and observed that, similar to caspase inhibition, this potently protected cells from death induced by CHK1i (Fig. 1d, Supplementary Fig. 1D). Similarly, deletion of pro-apoptotic BAX and BAK in Nalm6 pre-B ALL cells protected from apoptosis but is

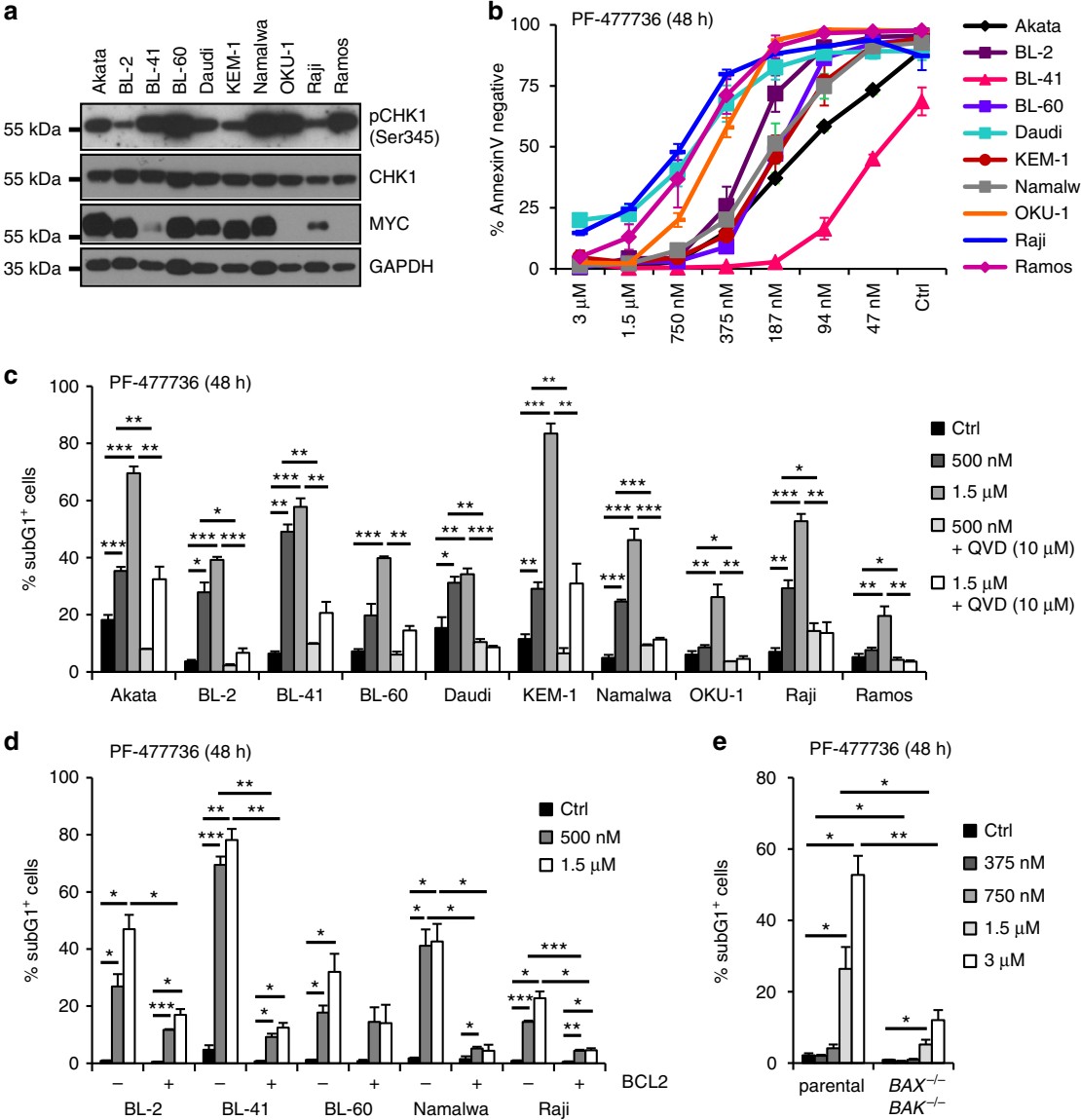

**Fig. 1** Inhibition of CHK1 results in BCL2-regulated cell death. **a** Western blot analysis showing the expression and activation status of CHK1 (pSer345) as well as MYC levels in ten different Burkitt lymphoma cells lines. **b** Dose response of Burkitt lymphoma cells lines to Chk1 inhibitor PF-477736. Survival was assessed using AnnexinV-staining and flow cytometry. **c** Burkitt lymphoma cells lines were treated for 48 h with the CHK1 inhibitor PF-477736 (CHK1i), either alone or in combination with the pan-caspase inhibitor QVD. Cells were processed for sub-G1 analysis using propidium iodide (PI) staining and flow cytometry. **d** Selected Burkitt lymphoma cell lines were transduced with a retrovirus enabling overexpression of anti-apoptotic BCL2 or an empty control vector. Cell death was assessed after 48 h of CHK1i treatment by sub-G1 analysis. **e** Nalm6, human B cell precursor acute lymphocytic leukemia (pre-B ALL) cells, were used to knock-out BAX and BAK using CRISPR/Cas9-based genome editing. The sub-G1 fraction of cells was determined by flow cytometry 48 h after CHK1i treatment. Bars represent means of $n = 3$/treatment ± S.E.M. *$p < 0.05$, **$p < 0.01$, ***$p < 0.001$ using unpaired Student´s $t$ test

associated with polyploidization, a phenomenon also seen in BCL2 overexpressing or QVD-treated cells (Fig. 1e, Supplementary Fig. 1E, F). Moreover, conditional RNAi-mediated depletion of CHK1 in BL2 Burkitt lymphoma and Nalm6 pre-B ALL cells coincided with the induction of apoptosis, as indicated by PARP1 cleavage, a marker of caspase activation (Supplementary Fig. 1B). Together, these data demonstrate that CHK1 is an essential survival factor for Burkitt lymphoma and pre-B ALL cells and that CHK1 inhibition activates BCL2-regulated and BAX/BAK-dependent apoptosis in these cells.

**Chk1 is essential for normal B cell development**. Considering CHK1 as a key target to treat B cell malignancies, we wondered about the consequences of CHK1 inhibition for normal B cells

and assessed its role in mouse B cell development. First, we explored *Chk1* expression by analyzing RNA-seq data generated from primary murine B cells, isolated at different developmental stages by cell sorting[37]. This revealed high levels of *Chk1* in strongly proliferating pro-B cells and large cycling pre-B cells from bone marrow as well as in mature B cells derived from spleen upon activation with anti-CD40 mAB. In contrast, mRNA levels were lower in non-proliferating small pre-B and IgM$^+$IgD$^-$ immature bone-marrow B cells and resting mature B cells from spleen. These findings were corroborated by qRT-PCR and western analysis (Fig. 2a, b). While these analyses showed a clear correlation of CHK1 protein levels with proliferation status, *Chk1* mRNA expression correlated only poorly with levels of *c-Myc* across B cell development (Fig. 2a), contrasting a previously reported interrelationship between these two genes[31, 38].

Next, $Chk1^{fl/-}$ mice were intercrossed with the B cell specific deleter-strain $Mb1$-$Cre$. $Chk1^{fl/-}$ $Mb1$-$Cre^+$ mice were born healthy and according to the expected Mendelian frequency. Strikingly, spleen size, weight and cellularity were significantly decreased in mice lacking $Chk1$ in B cells, indicating defects in normal B cell development (Fig. 2c, Supplementary Fig. 2A). Indeed, flow cytometric analysis revealed a pronounced decrease in the percentage of B220$^+$ B cells in the spleen (Fig. 2d, Supplementary Fig. 2A). Furthermore, the few remaining B220$^+$ cells lacked expression of mature B cell markers, IgM or IgD, indicating that these cells might represent dendritic cells.

Consistently with observations made in spleen, $Chk1^{fl/-}$ $Mb1$-$Cre^+$ mice also lacked differentiated B cells in the peripheral blood (Supplementary Fig. 2C) or lymph nodes (not shown). In line with our hypothesis, analysis of bone marrow revealed a dramatic decrease in immature B220$^+$ IgM$^-$ B cell progenitors in $Chk1^{fl/-}$ $Mb1$-$Cre^+$ mice. Most of the B220$^+$ cells were IgM$^-$ and c-Kit$^+$ but negative for CD25 indicative for a block in differentiation or cell loss at the pro- to pre-B cell transition (Fig. 2e; Supplementary Fig. 2B). Of note, this phenotype did not change over an observation period of one year, indicating that the developmental block imposed by $Chk1$-loss is permanent and the

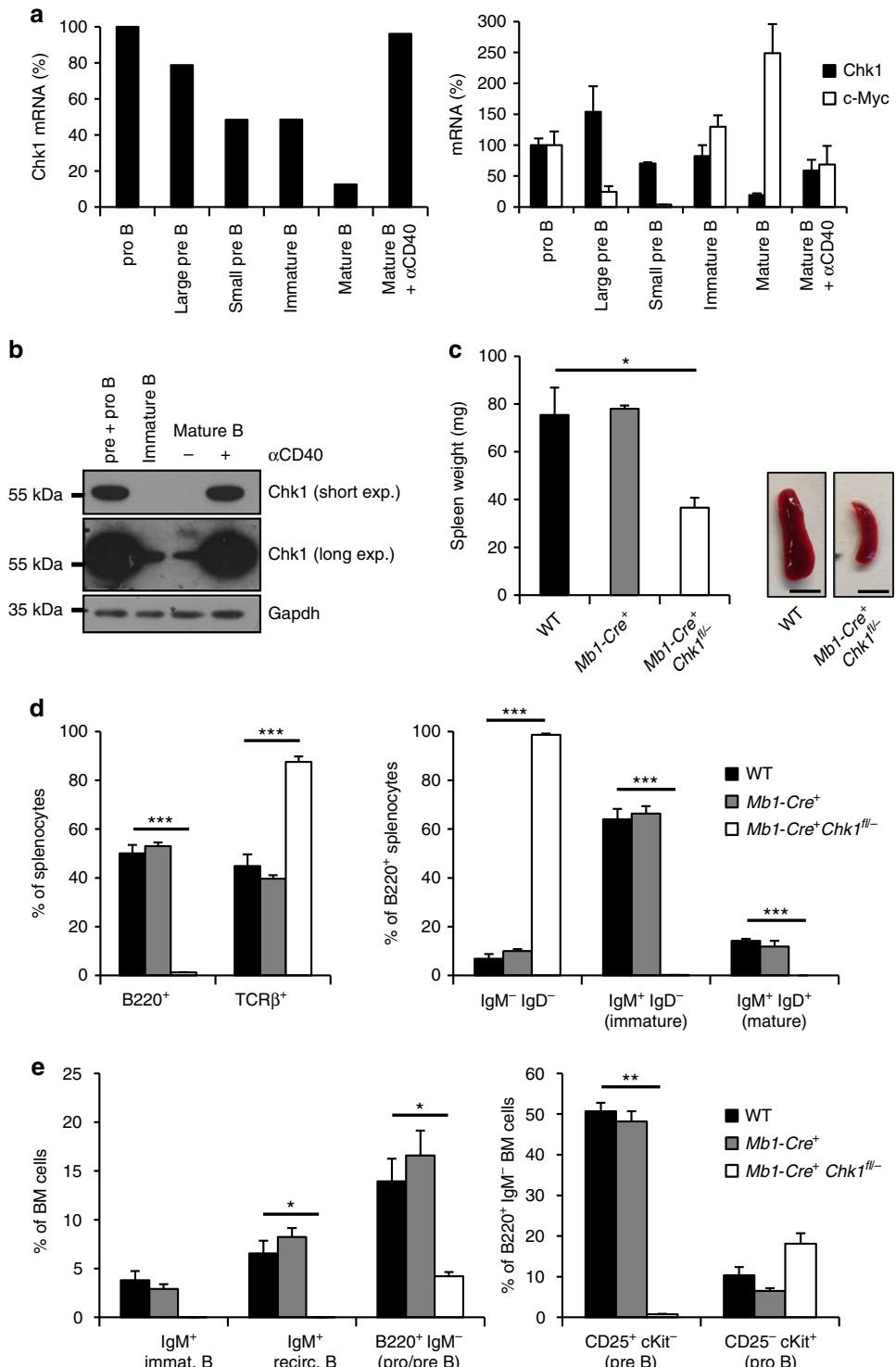

*Mb1-Cre* allele guarantees efficient target-gene deletion (Supplementary Fig. 3).

As inhibition of CHK1 can trigger DNA damage and apoptosis in cancer cell lines but may also prevent expansion of primary cells, we investigated the impact of a gene-dosage reduction and graded inhibitor treatment on bone marrow derived pro-B cells, isolated by cell sorting and cultured in the presence of Interleukin-7. Cytokine treatment promotes proliferation and differentiation of pro-B cells to the pre-B cell stage. Even though bone-marrow composition was comparable between wt and *Chk1* $^{+/-}$ mice (not shown), the proliferative capacity of pro-B cells isolated from *Chk1* $^{+/-}$ mice in culture was severely reduced and coincided with increased cell death in response to chemical CHK1 inhibition (Fig. 3a). Moreover, pre-B cells isolated from bone marrow from wild-type, but not those from *Vav-BCL2* transgenic mice[39], underwent apoptosis upon CHK1i treatment. Remarkably, resting mature B cells, but not resting mature T cells, isolated from the spleen of these mice also succumbed to cell death upon CHK1 inhibition. This death is apoptotic, as it was prevented effectively by BCL2 overexpression (Fig. 3b–d). Similar results were observed when using CHIR-124, although the effects on mature B cells were not as pronounced (Supplementary Fig. 4) and higher concentrations of inhibitor killed resting B and T cells in a BCL2-independent manner (not shown).

Considering apoptosis of pre-B cells lacking *Chk1* as the main cause for impaired B cell development, we tested whether B lymphopoiesis could be restored by blocking mitochondrial apoptosis. Interestingly, *Chk1* $^{fl/-}$ *Mb1-Cre* $^+$ mice overexpressing BCL2 showed the same phenotype in spleen and bone marrow (Fig. 4, Supplementary Fig. 2D, E), peripheral blood or lymph nodes (not shown) as *Mb1-Cre* $^+$*Chk1* $^{fl/-}$ mice. However, spleens of *Mb1-Cre* $^+$*Chk1* $^{fl/-}$*Vav-BCL2* mice were substantially smaller than *Vav-BCL2* spleens but were composed mainly of BCL2-transgenic T cells and macrophages (not shown). Of note, analysis of mice between 8 and 12 month of age revealed again that this block in B cell development persisted over time, even in the presence of the BCL2 transgene (Supplementary Fig. 3), suggesting that early B cells that might survive in the presence of BCL2 are unable to expand.

Taken together, this shows that *Chk1* is essential for normal B cell development and secures survival and proliferation of pre-B cells that also experience physiological DNA damage during B cell receptor rearrangement and, potentially, replication stress during their rapid expansion.

**Lymphomagenesis in mice depends on CHK1 expression levels.** Intrigued by these findings, we asked if *Chk1* is needed not only for B cell development and survival, but also for oncogene-driven B cell transformation. Mice transgenic for c-MYC, placed under the control of the *Igh* enhancer were shown to develop pro/pre-B or IgM $^+$ immature B cell lymphomas within their first year of life[40]. As it is known that high levels of MYC trigger replication stress-associated DNA damage but also indirectly increase CHK1 expression, we were addressing the question whether different levels of CHK1 affect the ability of c-MYC to drive B cell lymphomagenesis. First, we scrutinized the published interrelation between c-MYC and CHK1 expression[31]. Therefore, we isolated total splenocytes from *wt*, *Chk1* $^{+/-}$, *Eμ-MYC* and *Eμ-MYC Chk1* $^{+/-}$ mice for immunoblot analysis. CHK1 levels were indeed substantially increased in B cells from premalignant *Eμ-MYC* mice, when compared to wt controls, the latter showing approximately twice the levels found in *Chk1* $^{+/-}$ mice, as anticipated. Of note, *Eμ-MYC Chk1* $^{+/-}$ mice showed protein levels that were still higher than those found in wild-type animals lacking MYC overexpression, despite a discernible degree of variability (Fig. 5a). Together, this confirms a gene-dose relationship between MYC and CHK1 in *Eμ-MYC* transgenic B cells.

As pre-B cells lacking *Chk1* do not develop further (Fig. 2), this also precluded MYC-driven lymphomagenesis (Fig. 5b). Hence, we investigated whether *Chk1* haplo-insufficiency might affect tumor formation, as it was proposed to accelerate disease in other cancer models[23, 25, 41]. Therefore, we intercrossed *Chk1* $^{+/-}$ animals with *Eμ-MYC* mice and monitored disease onset over time. Remarkably, mice lacking one allele of *Chk1* showed a twofold increase in tumor latency, with a median occurrence of disease at 205 days vs. 106 days in controls (Fig. 5b). When we analyzed the tumors by flow cytometry, we did not observe major differences in their immune phenotype when discriminating pro/pre-B and immature B cell tumors using IgM expression as the only maturation marker, albeit, upon additional cell surface marker analysis, mixed pre-B/immature B (CD19 $^+$ AA4.1 $^+$ IgM $^+$) tumors were absent from mice lacking one allele of *Chk1* (Supplementary Fig. 5A, C). The significance or cause of this difference remains unclear. Upon development of disease, no difference in spleen weight was noted. Diseased *Eμ-MYC* and *Eμ-MYC Chk1* $^{+/-}$ mice developed similar splenomegaly whereas spleen weights from tumor-free, aged mice, sacrificed after one year were found unaffected (Supplementary Fig. 5B).

To interrogate if this phenomenon was restricted only to MYC-driven B cell lymphomas, we also explored a model of IR-induced thymic lymphomagenesis. Tumor formation in this model is known to involve repetitive DNA replication stress, subsequent to DNA damage, in hematopoietic stem and progenitor cells and compensatory proliferation to overcome repetitive myeloablation[42]. CHK1 protein levels were found increased in thymus and spleen 4 days after IR (Fig. 5c), a time point where compensatory proliferation is at its peak[42], suggesting that CHK1 is engaged under these conditions to control replication fidelity. Similar to our findings in the MYC model, IR-driven T cell lymphomas developed significantly later in *Chk1* $^{+/-}$ mice, suggesting a broader applicability of this phenomenon (Fig. 5d).

**Fig. 2** Chk1 is essential for early B cell development. **a** Relative *Chk1* mRNA levels during B cell development were assessed by analyzing RNA-sequencing data (left) and validated by real-time qPCR (right). RNA-Seq analysis was performed using pooled RNA samples derived from FACS-sorted bone marrow pro-B (B220 $^+$ μHC $^-$ ckit $^+$), large pre-B (B220 $^+$ μHC $^-$ CD25 $^+$ FSC $^{high}$), small pre-B (B220 $^+$ μHC $^-$ CD25 $^+$ FSC $^{low}$) and immature B cells (B220 $^+$ μHC $^+$ κHC $^-$) as well as splenic mature B cells (B220 $^+$) left untreated or treated with 1 μg/ml anti-CD40 antibody for 48 h that were isolated from wild-type C57Bl/6N mice (*n* = 4). Bars depict the relative number of sequence reads, normalized to the pro-B cell stage (arbitrary set to 100%). Real-time qPCR: analysis was performed using RNA samples derived from FACS-sorted bone marrow pro B (B220 $^+$ CD19 $^+$ AA4.1 $^+$ μHC $^-$ CD25 $^-$ ckit $^+$), large pre B (B220 $^+$ CD19 $^+$ AA4.1 $^+$ μHC $^-$ CD25 $^+$ ckit $^-$ FSC $^{high}$), small pre B (B220 $^+$ CD19 $^+$ AA4.1 $^+$ μHC $^-$ CD25 $^+$ ckit $^-$ FSC $^{low}$) and immature B cells (B220 $^+$ CD19 $^+$ AA4.1 $^+$ μHC $^+$) as well as splenic mature B cells (B220 $^+$ CD19 $^+$) left untreated or treated with 1 μg/ml anti-CD40 antibody for 48 h that were isolated from wild-type C57Bl/6N mice (*n* = 3). **b** Western analysis showing CHK1 protein expression throughout B cell development. Samples from three C57Bl/6N mice were pooled for analysis. BM pre + pro B cells: B220 $^+$ CD19 $^+$ AA4.1 $^+$ IgM $^-$, BM immature B cells: B220 $^+$ CD19 $^+$ AA4.1 $^-$ IgM $^+$, mature B cells (B220 $^+$ CD19 $^+$) from spleen were lysed immediately or treated with 1 μg/ml anti-CD40 antibody for 48 h. **c** Spleen weight and representative pictures (scale bar = 5 mm) of spleens of 6–8 weeks old mice of the indicated genotypes. **d** Quantification of flow cytometric analyses defining TCRβ $^+$ T and B220 $^+$ B cell populations in spleens of the indicated genotypes. **e** Flow cytometric analysis of bone marrow resident B cells in mice of the indicated genotypes. Bars in **c–e** represent means of *n* = 5 WT, *n* = 3 *Mb1-Cre* $^+$, *n* = 5 *Mb1-Cre* $^+$ *Chk1* $^{fl/-}$ ± S.E.M. \*p < 0.05, \*\*p < 0.01, \*\*\*p < 0.001 using unpaired Student´s *t* test

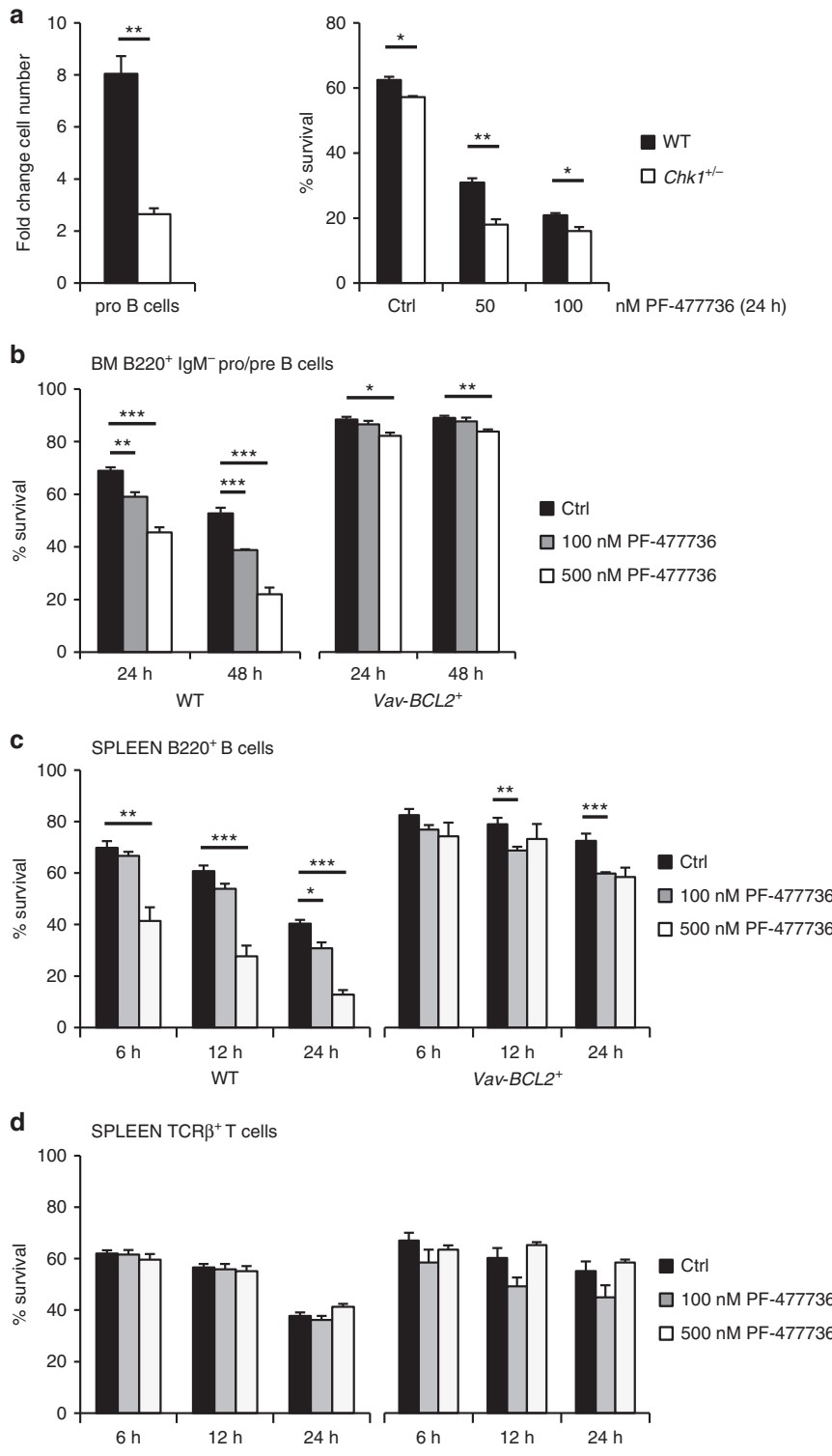

**Fig. 3** Overexpression of anti-apoptotic BCL2 blocks cell death by CHK1i in vitro **a** Pro B cells (NK1.1⁻ B220⁺ CD19⁺ AA4.1⁺ μHC⁻ CD25⁻ ckit⁺) were isolated from wild-type and *Chk1*⁺/⁻ C57Bl/6N mice and expanded for four days in IL-7 containing medium ($n = 3$/genotype). At day 4 cells were counted to assess cell expansion (left) and treated for additional 24 h with graded doses of the CHK1 inhibitor PF-477736 (right). Survival was assessed using AnnexinV-staining and flow cytometry. **b** FACS-sorted wild-type (WT) and BCL2 transgenic (*Vav-BCL2*⁺) bone marrow derived B220⁺ IgM⁻ pro/pre B cells were treated for 24 h and 48 h with PF-477736 or solvent control (DMSO). Cells were processed for survival analysis using a flow cytometer and AnnexinV/7AAD staining. **c**, **d** Total splenocytes were cultured for the indicated time in the presence of PF-477736 or solvent control (DMSO) and processed for survival analysis using B220/TCRβ/AnnexinV-staining and flow cytometry. **c** Quantification of B220⁺ AnnexinV⁻ cells. **d** Quantification of TCRβ⁺ AnnexinV⁻ cells. Bars in **b–d** represent means of $n = 3$ animals/genotype ± S.E.M. *$p < 0.05$, **$p < 0.01$, ***$p < 0.001$ using unpaired Student´s t test

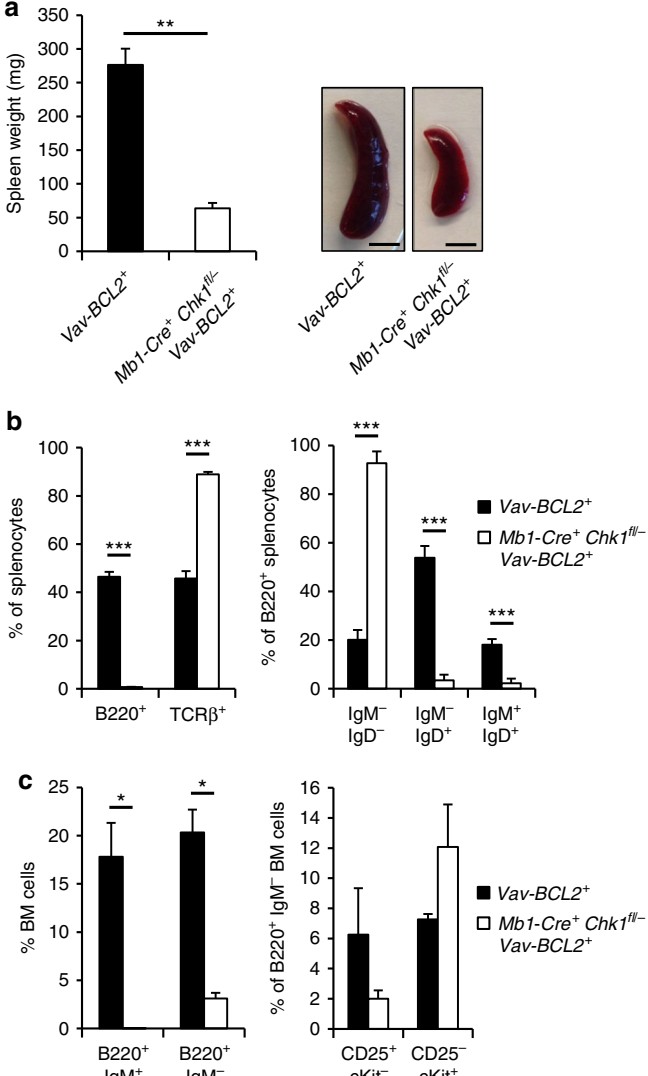

**Fig. 4** Inhibition of BCL2-regulated cell death fails to restore B cell development in the absence of CHK1. **a** Spleen weight and representative pictures (scale bar = 5 mm) of spleens of 6–8 weeks old mice of the indicated genotypes. **b** Relative distribution of splenic B cells (B220⁺) and T cells (TCRβ⁺) as well as different B cell subsets based on IgM and IgD expression of the indicated genotypes is shown. **c** Distribution of bone marrow B cell subsets. Bars represent means of $n = 4$ Vav-BCL2⁺ and $n = 5$ Mb1-Cre⁺ Chk1$^{fl/-}$ Vav-BCL2⁺ animals ± S.E.M. *$p < 0.05$, **$p < 0.01$, ***$p < 0.001$ using unpaired Student´s $t$ test

Together, these findings suggest that at least in lymphocytes, CHK1 expression levels are decisive for the kinetics of transformation.

***Chk1$^{+/-}$ B cells expressing MYC show increased DNA damage.*** The high proliferation rate of pre-B cells overexpressing MYC leads to malignant transformation only once initial anti-proliferative senescence-like or cell death-inducing responses are overcome[43]. Therefore, we reasoned that *Chk1* haploinsufficient cells might suffer a disadvantage in coping with replication stress caused by MYC overexpression. This might lead to increased cell death or might be balanced by anti-proliferative responses, ultimately delaying disease onset. However, when we analyzed B cells from premalignant *Eμ-MYC* and *Eμ-MYC Chk1$^{+/-}$* mice ex vivo, we could not detect significant differences in

spontaneous cell death of sorted splenic or bone-marrow derived B220⁺ IgM⁻ B cells that died rapidly in culture (Supplementary Fig. 6). Similar findings were made when we cultivated total splenocytes, to slow down ex vivo cell death rates, and assessed apoptosis by Annexin-V-staining over time (Supplementary Fig. 7A, B). Finally, we used BrdU incorporation to assess S-phase proficiency in premalignant mice. Again no difference was observed between the genotypes interrogated at different developmental stages (Supplementary Fig. 7C).

As mainly early pro/pre-B cells and IgM⁺ immature B cells are pushed to transform in *Eμ-MYC* transgenic mice, we next sorted these cells for western analysis to look by biochemical means for evidence of increased replication stress-driven DNA damage or apoptosis rates. Notably, premalignant B220⁺ IgM⁻ bone marrow B cells from *Eμ-MYC Chk1$^{+/-}$* mice seemed to differ from their *Chk1*-proficient counterparts in their ability to deal with MYC-driven replication stress, as indicated by increased levels of γH2A.X, as well as cleaved PARP1, a well-known caspase-3 substrate during apoptosis. Consistently, increased levels of γH2A.X in S-phase were observed in premalignant *Chk1$^{+/-}$* pre-B cells overexpressing MYC (Fig. 6a, b). A similar trend was noted in the spleen from these animals, albeit the effects were clearly not as pronounced, limiting the strength of our conclusion to bone marrow pre-B cells. Yet, the same phenomenon was again clearly visible in cell death-refractory pre-B cell lines generated from 2-weeks old *Eμ-MYC Vav-BCL2* mice, either haploinsufficient for *Chk1* (Fig. 6c) or treated with CHK1i (Fig. 6d). These cells showed increased levels of γH2A.X staining by flow cytometry that was also prominently enriched in S-phase cells. Of note, phosphorylated H2A.X was confirmed upon CHK1i treatment by western that coincided with p53 stabilization (Fig. 7a).

Together these findings suggest that *Chk1* haploinsufficiency, similar to CHK1 inhibitor treatment, increases replication-stress associated DNA damage and increases susceptibility to apoptosis in premalignant cells that ultimately delays lymphocyte transformation in mice (Fig. 8a). Alternatively, DNA damage may also arise in cells that attempt to proliferate after failing cytokinesis due to loss of one allele of *Chk1*[44], warranting future investigations into the exact molecular basis of this phenomenon.

**CHK1 is needed for the growth of apoptosis-resistant B cells.** Together, our observations suggest that *Chk1* is essential for B cell survival, transformation and tumor cell maintenance in mice. Yet the notion why BCL2 overexpression can rescue primary mouse B cells and human cancer cells from apoptosis in vitro but fails to restore murine B cell development in vivo remains unexplained. We reasoned that in the absence of cell death the accumulation of DNA damage (primary/premalignant B cells), mitotic errors or polyploidization (tumor cell lines) might ultimately trigger cell cycle arrest or senescence that might explain our findings in vivo. To this end we performed proliferation assays with apoptosis-resistant BCL2 overexpressing MYC-transgenic mouse B cells (Fig. 7a), pre-B ALL cells lacking BAX/BAK (Fig. 7b), as well as BCL2 overexpressing Burkitt lymphoma cell lines (Fig. 7c). Strikingly, all cell types tested responded with severely reduced growth in population-doubling assays. In p53-proficient cells, this was associated with p53 stabilization and induction of p21, suggesting activation of a canonical DDR in response to DNA damage induced by CHK1 inhibition, reducing or inhibiting proliferation of cell death defective B cells. p53-impaired Burkitt lymphoma cells (all but BL2 and OKU-1 http://p53.free.fr/Database/p53_mutation_HB.html) did not increase in cell number, but as can be seen in Supplementary Fig. 1F, developed severe polyploidy/aneuploidy profiles when cell death was prevented by BCL2 overexpression. This feature was also seen in Nalm6 cells

lacking BAX/BAK, despite the induction of p53 (Supplementary Fig. 1F). The latter observation can be reconciled with a role of CHK1 in controlling Aurora B kinase localization for cytokinesis[44], albeit potential off target effects of inhibitor treatment cannot be excluded.

Together, these findings suggest that targeting CHK1 is a valid strategy to hit hard-to-treat blood cancer with high apoptotic threshold or low apoptotic priming[45]. A limitation of this strategy, however, might be that, in the absence of functional p53, or reduced cell cycle check-point proficiency, BCL2-controlled apoptosis resistance can trigger aneuploidy tolerance, increasing the chances for treatment failure (Fig. 8b).

## Discussion

Here, we show that *Chk1* is essential for normal B lymphopoiesis and that CHK1 expression levels affect transformation of lymphocytes induced by MYC overexpression or irradiation damage, in line with a core pro-survival role for CHK1 in murine lymphomas. Consistently, inhibition of CHK1 effectively kills human Burkitt lymphoma and pre-B ALL cells but also affects primary mouse B cells. Cell death caused by CHK1i or B cell specific deletion involves intrinsic apoptosis, as overexpression of BCL2 is highly protective. Yet, blocking apoptosis alone is insufficient to allow the subsequent outgrowth of primary B cells lacking CHK1 in vivo or B lymphoma lines in vitro, as these cells trigger p53-dependent cell cycle arrest in response to the accumulating DNA damage. As such, while the effects on primary B cells may potentially affect immune cell homeostasis in patients, the spread of potentially drug-resistant tumors expressing high levels of anti-apoptotic BCL2 family proteins may still be limited effectively by secondary anti-proliferative effects initiated upon CHK1 inhibition, making this kinase a highly attractive therapeutic target, as long as p53 function is not impaired. Yet, in the absence of functional p53 or reduced checkpoint-proficiency, CHK1i promotes polyploidization and apoptosis resistance will increase the risk for aneuploidy tolerance, driving the selection of more aggressive disease.

While early findings using *Chk1* haploinsufficient mice suggested a tumor suppressive function in response to, e.g., aberrant *Wnt1* expression in the mammary gland or in the context of *Chk2*-deficiency[23, 25, 41], detailed analysis suggested a delicate gene-dose relationship in cancer formation. While haploinsufficiency of *Chk1* facilitated also intestinal cancer, skin cancer progression or synergized with p53 haploinsufficiency in the mammary gland, deletion of *Chk1* prevented tumor formation in either model system[24, 41, 46]. Together, these studies suggest that reduced levels of CHK1 can facilitate tumor formation or progression, but total loss is incompatible with tumor cell formation. Our data, however, suggest that even a reduction of CHK1 expression can be tumor suppressive, at least in the context of replication stress-driven tumors, modeled here by oncogenic MYC or irradiation damage. As tumor formation in these models involves either continuous or repeated replication stress and DNA damage during compensatory proliferation, respectively[35, 42], we propose that CHK1 exerts "oncogenic" potential under these conditions in lymphocytes, while it may well act as a haploinsufficient tumor suppressor in the mammary gland, the skin or on a Chk2-deficient background[24, 46]. We propose that this difference may be due to the increased cell death vulnerability of

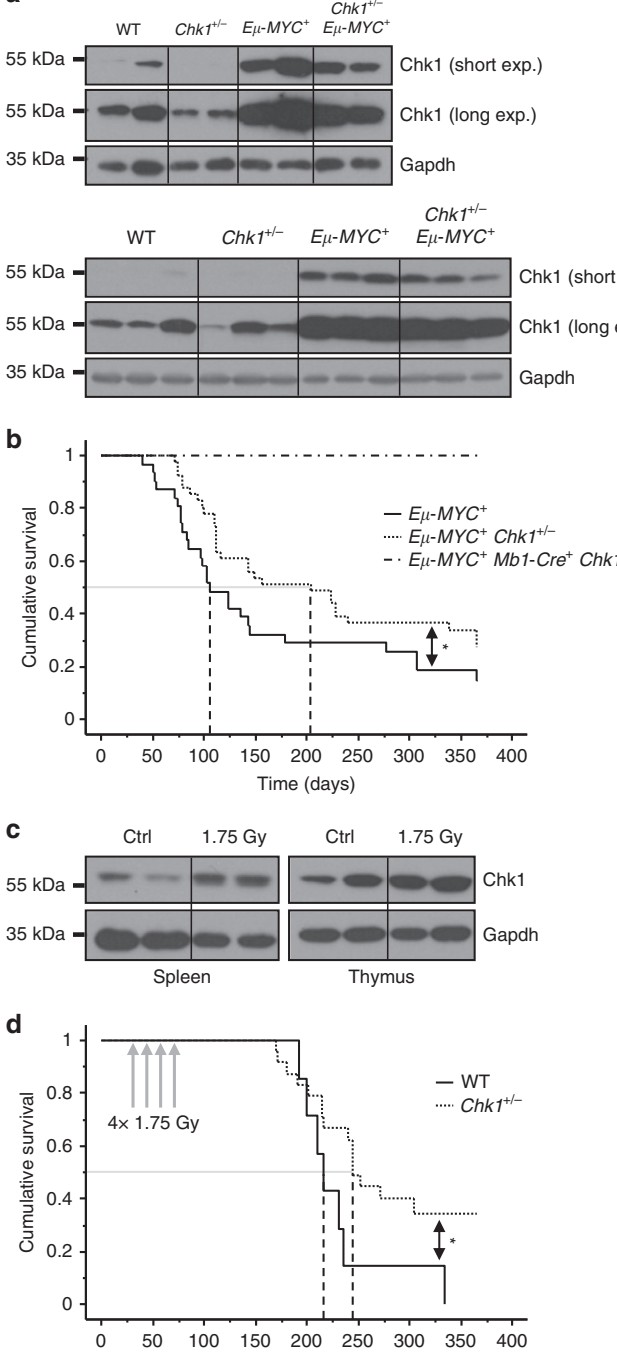

**Fig. 5** Successful lymphocyte transformation depends on CHK1 expression levels. **a** Total splenocytes of the indicated genotypes were processed for immunoblotting using the indicated antibodies to assess the impact of MYC on CHK1 protein levels. Each lane represents an independent biological replicate. Two independent experiments are shown ($n = 5$/genotype in total) **b** Kaplan–Meier plot analysis showing lymphoma-free survival of mice of the indicated genotypes over an observation period of one year. $E\mu$-MYC$^+$ $n = 31$, median survival 106 days, $E\mu$-MYC$^+$ Chk1$^{+/-}$ $n = 41$, median survival 205 days. Logrank (Mantel–Cox) $p = 0.056$ ($\chi^2 = 3.632$), Breslow–Gehan–Wilcoxon $p = 0.0258$ ($\chi^2 = 4.967$). **c** Wild-type C57Bl/6N mice were either left untreated or irradiated with a single dose of 1.75 Gy. Four days after irradiation organs were harvested and processed for western analysis. Each lane represents an independent biological replicate. **d** Kaplan–Meier analysis showing lymphoma-free survival of WT and Chk1$^{+/-}$ mice exposed to a fractionated irradiation protocol (4× 1.75 Gy) at 4 weeks of age. Median tumor onset: WT ($n = 7$) 217 days, Chk1$^{+/-}$ ($n = 24$) 245 days. Logrank (Mantel–Cox) $p = 0.0399$ ($\chi^2 = 4.221$)

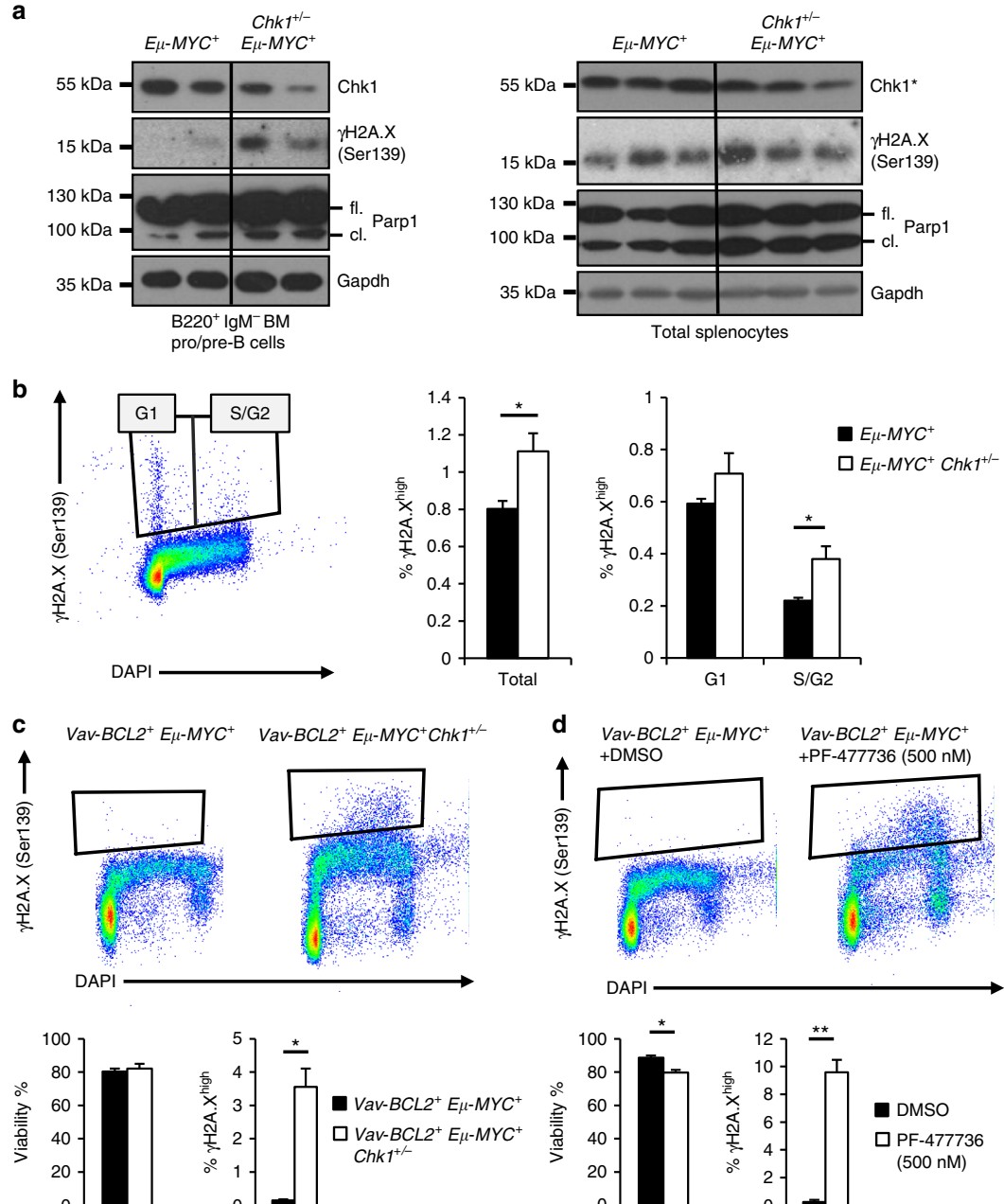

**Fig. 6** Haploinsufficiency in *Chk1* predisposes to DNA damage in response to MYC overexpression. **a** B220+ IgM− pro/pre B cells FACS sorted from the bone marrow (left) or total splenocytes of mice of the indicated genotypes were processed immediately for immunoblotting using the indicated antibodies. * Indicates that the same CHK1 western is also shown in Fig. 5a, bottom right, lanes 7–12. Membranes were reprobed for PARP1 and γH2A.X here **b** FACS sorted bone marrow B220+ IgM− pro/pre B cells were fixed in ethanol and then processed for intracellular γH2A.X staining. Bars represent means of n = 4–5 (*Eμ-MYC+/Eμ-MYC+ Chk1+/−*) animals ± S.E.M. **c, d** γH2A.X staining of pre B cell lines generated from the bone marrow of 2-weeks-old VavBCL2+ Eμ-MYC+ Chk1+/+ or VavBCL2+ Eμ-MYC+ Chk1+/− mice. **c** Steady state γH2A.X -levels **d** γH2A.X -levels of VavBCL2+ Eμ-MYC+ cells treated with 500 nM PF-477736 (CHK1i) or solvent control (DMSO) for 24 h (n = 3). Bars in **c** and **d** represent means of n = 3/genotype or treatment ± S.E.M. *p < 0.05, **p < 0.01, ***p < 0.001 using unpaired Student´s t test. Abbreviations: fl. full length, cl. cleaved fragment of PARP1

lymphocytes facing DNA damage that may reduce the chances of clonogenic outgrowth.

Consistent with the above, B cell lymphomagenesis was recently shown to be abolished in Atr-hypomorphic *Seckel* mutant mice (*Atr^{S/S}*), displaying reduced levels of the key kinase needed to activate CHK1 in response to replication stress and ssDNA breaks[35]. In *Eμ-MYC Atr^{S/S}* mice, pre-tumoral expansion of white blood cells was clearly prevented, further supporting the hypothesis that inhibiting CHK1 activity can prevent transformation at early stages. However, as *Eμ-MYC Atr^{S/S}* mice display

drastically reduced lifespan and die of pleiotropic disease, this study failed to unambiguously separate the consequences of impaired ATR from impaired downstream CHK1 signaling for cancer formation[35]. However, using B cell specific *Chk1* knock-out mice we could show that *Eμ-MYC* driven tumors highly depend on CHK1 as none of our *Mb1-Cre Chk1^{fl/−}* mice expressing the MYC transgene developed cancer over an observation period of one year and disease onset was found delayed in *Chk1+/−* mice. Therefore, loss of *Chk1* in B cells causes similar effects than impairing *Atr* function as both, *Atr^{S/S}* mice and *Eμ-*

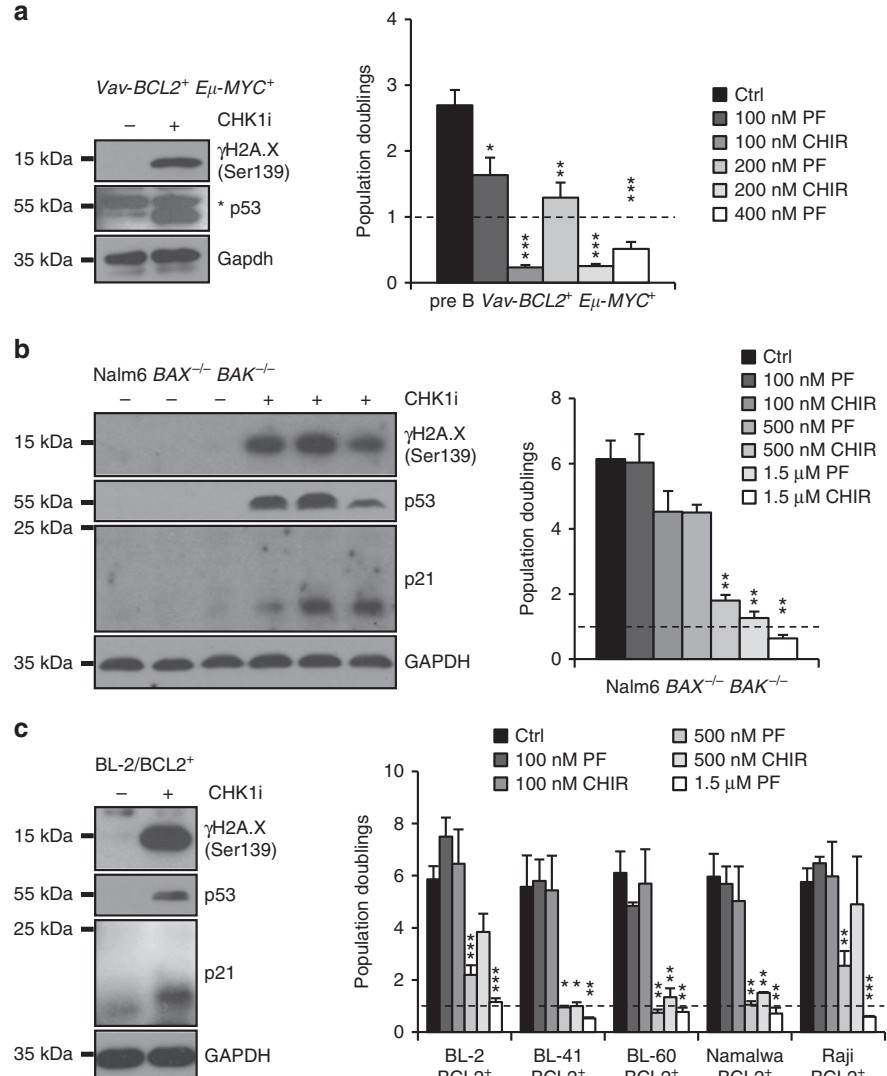

**Fig. 7** CHK1 inhibition induces cell cycle arrest in apoptosis-resistant cells. **a** Murine primary pre B cell lines as described in Fig. 6c, d. Left: western blot analysis ± CHK1 inhibition (500 nM PF-477736). Right: population doublings 48 h after treatment with two different CHK1 inhibitors PF-477736 or CHIR-124 or solvent control (DMSO) ($n = 3$). **b** Nalm6 $BAX^{-/-}BAK^{-/-}$ cells were treated (left) with PF-477736 (1.5 μM) or solvent control (DMSO) and processed for western blot. Right: population doublings as described in **a** ($n = 4$). **c** The BCL2 overexpressing BL-2 cell line was treated with (left) PF-477736 (1.5 μM) or solvent control (DMSO) and processed for western blot. Right: population doublings as described in **a** from indicated BCL2 overexpressing Burkitt lymphoma cell lines ($n = 3$). * p53: unspecific band. Bars represent means ± S.E.M., *$p < 0.05$, **$p < 0.01$, *** $p < 0.001$ using unpaired Student´s $t$ test

*MYC Mb1cre⁺ Chk1ᶠˡ/⁻* mice do not develop disease. Thus, interfering with the RSR pathway *per se* prohibits cancer formation at least in MYC driven malignancies in mice. Along similar lines, the relation between transformation and CHK1 levels was also assessed by others in a mouse model with increased expression. *Chk1ᵗᵍ* MEFs were resistant to UCN-01, hydroxyl urea and aphidicolin treatment and could even compensate the lower levels of ATR kinase in the *Atr-Seckel* mouse model, resulting in an enhanced lifespan of *Chk1ᵗᵍ Atrˢ/ˢ* mice. Additionally, *Chk1ᵗᵍ* MEFs were more prone to transform when transduced with Ha-Rasⱽ¹² plus E1A[47].

As the ATM–CHK2–p53 axis is often mutated in human cancers the ATR–CHK1–Cdc25 pathway is the only route left for cancer cells to survive in an environment of proliferation-induced replicative stress or exogenously inflicted DNA damage. Therefore, weakening the RSR by interfering with the ATR/CHK1 signaling axis seems to be a suitable strategy not only to prevent tumor formation in experimental models, supporting an

essential role of ATR and CHK1, but also for killing established tumors displaying a high proliferative index, such as *Eμ-MYC* lymphoma, Burkitt lymphoma or AML[30–33]. Of note, synthetic lethality of CHK1 (AZD7762) and ATR (VE-821 and VX-970) inhibitors was recently described in U2OS, VH-10, and MCF-7 cells, as well as in mouse xenotransplant models[36]. Overall these studies suggest that ATR/CHK1 inhibition triggers tumor cell death but it has not been addressed by what mechanism these cells die[32]. Based on our observations and findings by others documenting caspase-3 activation in cancer cells exposed to CHK1 inhibitors[33, 36], we conclude that BCL2-regulated apoptosis is elicited, as primary B cells or Burkitt lymphomas overexpressing BCL2 or pre-B ALL cells lacking BAX/BAK proved cell death resistant *ex vivo*, confirming mitochondrial apoptosis as a key event.

Depleting *Chk1* in B cells using *Mb1-Cre* drastically reduced the number and percentage of B cells in the bone marrow, spleen, lymph nodes and peripheral blood. These findings are

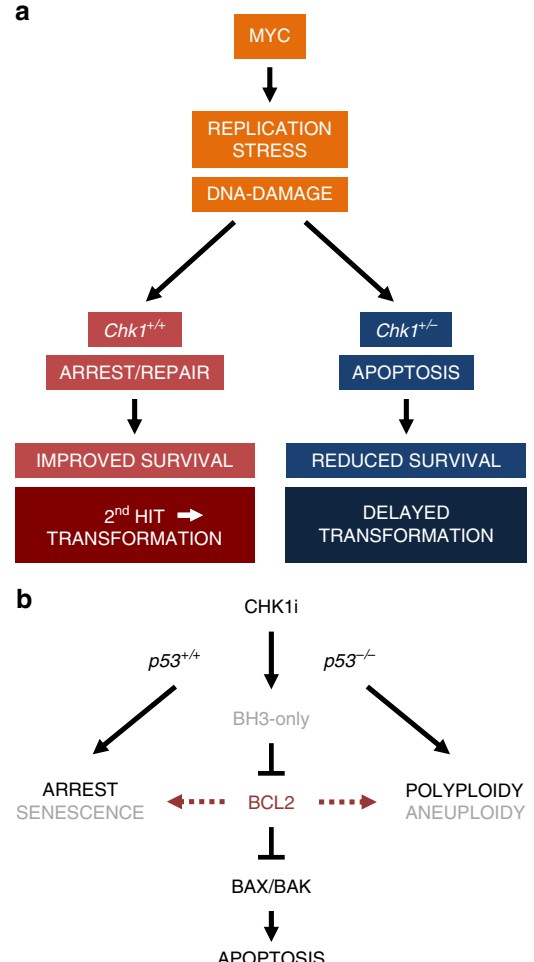

**Fig. 8** Proposed model of CHK1 as a facilitator of transformation and drug target. **a** Lymphocytes rely on CHK1 function to control replication fidelity to avoid DNA damage. In response to oncogenic stress, e.g., the one caused by MYC, replication stress and subsequent DNA damage increase the dependence on CHK1 for successful transformation that, in this model, depends on additional genetic alterations (2nd hit). A reduced CHK1 expression level curtails the cell´s ability to effectively deal with oncogene-driven replication stress, leading to increased DNA damage and apoptosis rates, thereby delaying transformation. **b** CHK1 inhibitors induce cell death in Burkitt lymphoma and ALL cells by engaging the BCL2-regulated BAX/BAK-dependent apoptosis pathway, kick-started by yet to be defined BH3-only proteins. Cell fate in response to inhibitor treatment is fine-tuned by p53 status and intrinsic apoptosis thresholds, controlled by the BCL2 family. Impaired cell death, e.g., due to BCL2 overexpression can deviate the cellular response to p21-mediated cell death, and, potentially senescence, when p53 is functional. Yet, in the absence of p53, cell death and cell cycle incompetence facilitates polyploidy of cancer cells exposed to CHK1 inhibitors. This can drive the selection of complex aneuploidy karyotypes and treatment failure

reminiscent to what has been observed upon T cell specific depletion of *Chk1* using *Lck-Cre* that arrests T cell development at the early double negative (DN)-stage where thymocytes that manage to express a functional pre-T cell receptor (TCR) expand before maturing further[48]. Of note, *Lck-Cre Chk1fl/fl* mice died prematurely. Despite the fact that this observation might be indicative for emergence of malignant disease, in line with a tumor suppressor role of *Chk1*, the cause of death was not clarified[48] and could eventually be due to T cell deficiency. In contrast, mice

lacking *Chk1* in B cells do not develop such a phenotype in our facility, at least over an observation period of one year. Similar to *Lck-Cre Chk1fl/fl* mice, *Mb1-Cre Chk1fl/−* mice failed to restore B lymphopoiesis when cell death was blocked by BCL2 overexpression. Based on our observations in cell lines, we postulate that under such conditions CHK1-deficient lymphocytes fail to expand, as they initiate cell cycle arrest in response to accumulating DNA damage. As we failed to note accumulation of B cell precursors or the generation of mature B cells over an observation period of one year it remains possible that these progenitor cells are eventually cleared by other cell death mechanisms than apoptosis.

At the moment it remains uncertain what consequences systemic CHK1 inhibition will have. *Chk1+/−* mice appear normal but a fraction of aged mice develop an anemic phenotype, consistent with a lower level of CHK1 mRNA in patients suffering from refractory anemia[49]. The role of CHK1 in normal erythropoiesis however remains unclear. Based on our preliminary findings, B cells appear more vulnerable than T cells to CHK1 inhibition in vitro, but this needs to be investigated in more detail. Of note, hepatocytes appear to be unaffected by conditional *Chk1* deletion driven by *Ah-Cre* activated upon beta-naphtoflavone injection[50]. This might be explained by the fact that these cells do show a very low proliferative index and based on this analogy, mature resting lymphocytes might survive *Chk1* depletion or inhibition in the absence of antigenic challenge, yet immunity, depending on the rapid expansion of antigen-specific lymphocytes, might become severely compromised in patients on CHK1i treatment. In support of this, gastrointestinal epithelial cells and the developing mammary gland are highly proliferative and susceptible to *Chk1* ablation. As a consequence the tissue is rapidly replaced by cells that arise from stem cells that have escaped *Chk1* deletion, confirming its essential role for cell survival in fast proliferating tissues[24, 50]. Because the tumor onset was significantly delayed in *Chk1* heterozygous *Eµ-MYC* mice and B cells showed a dose-dependent response to CHK1i ex vivo, we propose that there is a potential window of opportunity to treat CHK1i sensitive tumors without inducing severe side-effects that associate with complete loss of CHK1 activity.

It is interesting to note that the cell death observed upon *Chk1* loss in different tissues or early embryos cannot be restored by loss of p53 alone, or, as exemplified here, BCL2 overexpression blunting all mitochondrial apoptosis, being superior to loss of p53 in this regard. Yet, it is easy to imagine that in the context of BCL2 overexpression, cells can mount a p53-dependent cell cycle arrest, explaining the lack of B cells in *Mb1-Cre Chk1fl/−* mice carrying a BCL2 transgene. In contrast, upon loss of p53, these cells may be unable to arrest in the presence of accumulating DNA damage and undergo mitotic catastrophe. Hence, it will be interesting to explore the consequences of BCL2 overexpression in the absence of p53 in more detail in this context. Of note, many of the Burkitt lymphomas tested here harbor p53 mutations, demonstrating the potential of CHK1i to limit growth of cancer cells lacking functional p53. In the context of BCL2 overexpression, however, these cells seem to undergo enhanced polyploidization, as a consequence of failed cytokinesis, a process that is co-regulated by CHK1[44]. Usually, extra centrosomes present in such cells would engage the PIDDosome multiprotein complex to trigger p53 activation for cell cycle arrest[51]. However, when reentering mitosis in the presence of extra centrosomes such polyploid cells will suffer severe genomic instability and develop aneuploidy, initially incompatible with cell survival but as such a potent driver of tumor evolution and drug resistance. Hence, it will be of significant interest to identify the relevant effectors mediating cell death and cell cycle arrest engaged in response to CHK1 inactivation in p53-proficient as well as p53-

deficient settings. This will help to identify possible mediators of drug resistance and aneuploidy tolerance, potentially undermining this highly promising approach of cancer therapy.

## Methods

**Mouse models.** Animal experiments were performed in accordance with Austrian legislation (BMWF: 66-011/0106-WF/3b/2015). The generation and genotyping of *Chk1*[fl/fl], *Vav-BCL2*, *Mb1-Cre*, and *Eμ-MYC* mice have been described[23, 39, 40, 52]. All mice were back-crossed on C57BL/6N. To induce thymic lymphomas, mice at the age of 4 weeks were irradiated with 1.75 Gy in a linear accelerator once per week for 4 weeks, starting at the age of four weeks.

**Cell culture, virus production and reagents.** Cells were cultured at 37 °C in a humidified atmosphere containing 5% $CO_2$, except for mouse pre-B cells (7.5% $CO_2$). Human Burkitt lymphoma, Nalm6 cells and murine pre-B cell lines were cultured in RPMI-1640 medium (Sigma-Aldrich, R0883) supplemented with 10% FCS (Gibco, 10270-106), 2 mM L-glutamine (Sigma, G7513), 100 U/ml penicillin and 100 μg/ml streptomycin (Sigma, P0781) and 50 μM 2-mercaptoethanol (Sigma, M3148). For murine pre-B cells and primary bone marrow culture, 1 mM sodium pyruvate (Gibco, 11360-039), 1× non-essential amino acids (Gibco, 11140-35) and IL-7 (supernatant of IL-7 expressing J558L cells) was added in addition. Primary FACS-sorted murine pro B cells were cultured in DMEM medium (Sigma, D5671), supplemented with 10% FCS, 100 U/ml penicillin and 100 μg/ml streptomycin, 2 mM L-glutamine, 10 mM Hepes (LONZA, BE17-737E), 1 mM sodium pyruvate, 1× non-essential amino acids and 50 μM 2-mercaptoethanol. Human embryonic kidney cells (HEK293T) were grown in DMEM medium, supplemented with 10% FCS, 100 U/ml penicillin and 100 μg/ml streptomycin and 2 mM L-glutamine. Reagents: PF-477736 (Selleckchem S2904), CHIR-124 (Selleckchem S2683), QVD (SML0063, Sigma), DMSO (D5879, Sigma), Doxycycline (D9891, Sigma), anti-mouse CD40 (102908, biolegend, HM40-3).

**Viral transduction, RNAi and CRISPR/Cas9 genome editing.** Burkitt lymphoma cells were transduced with a pMIG-based retrovirus encoding BCL2 and a pur-omycin resistance marker (pMIG-BCL2) by repeated spin infection at 37 °C (3 × 30′ 800 g), followed by selection for 2–3 days with puromycin 2 μg/ml. For generating lentiviral particles, HEK293 cells were plated at 50% density. Two hours before transfection media was exchanged and cells transfected with the viral packaging plasmids pSPAX2, the pseudotyping plasmid pVSV-G and the lentiviral vectors lentiCRISPRv2 or lentiCRISPRv4 (both Addgene) at a 1:1:2 ratio using calcium phosphate co-precipitation. One day after transfection media was exchanged and supernatant containing lentiviral particles were harvested at 48 h and 72 h. Virus-containing supernatant was sterile filtered, supplemented with 4 μg/ml Polybrene and added to Nalm6 cells for 2–3 days followed by selection using puromycin 4 μg/ml or blasticidin 5 μg/ml. The guideRNAs BAK (lentiCRISPRv4: GCCATGCTGGTAGACGTGTA; lentiCRISPRv2: GGCCATGCTGGTAGACGT GT) or BAX (lentiCRISPRv4: CAAGCGCATCGGGGACGAAC; lentiCRISPRv2: CGAGTGTCTCAAGCGCATCG) were selected using Crispr-design at http://crispr.mit.edu. Double stranded DNA-oligonulceotides were 5′ phosphorylated and cloned into the BsmBI linearized lentiCRISPR vectors. After selection cells were subjected to limiting dilution cloning. Cell lysates were subjected to immuno-blotting using antibodies for BAX (Cell Signaling 2772) or BAK (Cell Signaling 9814), respectively. To confirm the desired genetic modification, genomic DNA was isolated, the targeted exons amplified by PCR and subjected to DNA sequencing. CHK1 targeting short hairpin RNA (shRNA) encoding oligonuicleo-tides were cloned into HindIII-BglII digested pENTR-THT-III, a GATEWAY cloning compatible ENTR vector harboring a tetracycline-regulatable H1-RNA gene promoter. After sequence verification, the shRNA expression cassette was shuttled into a selectable puromycin resistance conferring as well as a TetR-GFP expressing lentiviral vector. Target cells were transduced with lenviral particles generated by transient transfection of lentiviral constructs with packaging and pseudotyping plasmids. Transduced target cells were selected using 2.5 μg/ml puromycin and induced using 1 μg/ml doxycycline. Target site in CHK1 5′ GAAGCAGTCGCAGTGAAGA.

**Intracellular staining and DNA-content analysis.** Cells were fixed in 1 ml ethanol (70 %) and stored at −20 °C for a minimum of 60 min. Prior antibody staining, cells were washed twice (800 g, 5′) with 2 ml PBS to remove ethanol, followed by 15′ incubation in TritonX100 (0.25%, Sigma) on ice for permeabilization. Cells were incubated 60′ with anti-Human/Mouse phospho-H2AX S139 mAb PerCP-eFluor 710, (clone CR55T33, ebioscience). Cells were washed in PBS 1% BSA. DAPI was used for DNA content analysis. Alternatively, cell cycle distribution was assessed after fixation in 70% ethanol and propidium iodide staining (4 μg/ml; Sigma).

**Antibodies, flow cytometry and cell sorting.** Flow-cytometric analysis or cell sorting of single cell suspensions generated from spleen or bone marrow was performed on an LSR-Fortessa or a FACS-Aria-III, respectively (both BD) and analyzed using FlowJo v10 software. Antibodies used (40 μl/sample): B220-FITC (biolegend RA3-6B2) 1:300, B220-APC/eF780 (ebioscience RA3-6B2) 1:200, B220-

PE (BD RA3-6B2) 1:300, B220-PerCP-Cy5.5 (biolegend RA3-6B2) 1:300, CD19-BV605 (biolegend 6D5) 1:400, IgM-APC (biolegend RMM-1) 1:300, IgM-FITC (BD II/41) 1:300, IgM-eF450 (ebioscience eb121-15F9) 1:200, IgD-PerCP-Cy5.5 (biolegend 11-26c2a) 1:400, TCRβ-BV605 (BD H57-597) 1:200, TCRβ-FITC (ebioscience H57-597) 1:300, CD4-APC/Cy7 (BD GK1.5) 1:400, cKit-PE/Cy7 (biolegend 2B8) 1:300, cKit-PerCP-Cy5.5 (biolegend 2B8) 1:300, cKit-BV421 (biolegend 2B8) 1:300, Mac1-APC (ebioscience M1/70) 1:300, CD25-PE (biolegend PC61) 1:500, CD93-PE/Cy7 (ebioscience AA4.1) 1:300, NK1.1-APC (biolegend PK136) 1:300, AnnexinV-FITC (biolegend Lot: B206041) 1:1800, AnnexinV-eF450 (ebioscience; Lot: E11738-1633) 1:1000. FITC-F(ab')$_2$ Fragment Goat Anti-Mouse IgM [1 μg/ml], μ Chain Specific (Jackson ImmunoResearch).

**Gene expression.** RNA was isolated from snap-frozen FACS-sorted or in vitro cultivated B cell subsets using the Qiagen RNeasy Mini Kit (74104) and the RNase-Free DNase Set (79254) according to the manufacturer's instructions. For each sample, hundred nanograms of RNA were reversed to cDNA using the iScript cDNA Synthesis Kit (1708890) according to the manufacturer's instructions. For qRT-PCR with the StepOne Plus System (Applied Biosystems), we used the 2x SYBR Green qPCR Master Mix from Biotool (B21203) according to the manu-facturer's instructions. Expression levels of CHK1 and c-MYC were normalized to the housekeeping gene HPRT. Primers used were as follows (5′–3′): Chk1_Fw CTCCATCAGCAAGGATCACC, Chk1_Rv ACGGTTTCTTCACTGGAACC c-Myc_Fw GCTGTTTGAAGGCTGGATTTC, c-Myc_Rv GATGAAATAGGGCTG TACGGAG, Hprt_Fw GTCATGCCGACCCGCAGTC Hprt_Rv GTCCTTCCAT AATAGTCCATGAGGAATAAAC.

**BrdU incorporation and viability assay.** Proliferation of immature (B220[+] IgM[−]) and mature (B220[+] IgM[+]D[+]) B cells was assessed by injecting 1 mg BrdU/mouse i.p. Four hours later primary cells were isolated and stained for BrdU incorporation using the BrdU/APC flow kit (BD, Vienna, Austria) according to manufacturer's recommendation. Cells were co-stained with Annexin-V-eF450 (1:1800 in Annexin-V binding buffer; BD) and 7-AAD (1 μg/ml, Sigma) and spontaneous cell death was analyzed by subsequent flow cytometric analysis.

**Immunoblotting.** Cells were lysed in 50 mM Tris pH 8.0, 150 mM NaCl, 0.5% NP-40, 50 mM NaF, 1 mM Na$_3$VO$_4$, 1 mM PMSF, one tablet protease inhibitors (EDTA free, Roche) per 10 ml and 30 μg/ml DNaseI (Sigma-Aldrich) and analyzed by immunoblotting. For detection of proteins by chemoluminescence (Advansta, K-12049-D50) a mouse anti-CHK1 (CS 2360, 2G1D5) 1:750, mouse anti-CHK1Ser345 (CS 2348) 1:500, mouse anti-MYC cMYC (Y69 Abcam 32072) 1:1000, rabbit anti-GAPDH (CS 2118, 14C10) 1:5000, rabbit anti-γH2AX (CS 2577) 1:1000, rabbit anti-p53 (Santa Cruz sc-6243) 1:200, mouse anti-p21 (BD clone SX118) 1:500 or rabbit anti-PARP1 (CS 9542) 1:1000 were used (5 ml/ membrane). Goat anti-rabbit Ig/HRP (Dako, P0448) or rabbit anti-mouse Ig/HRP (Dako, P0161) were used as secondary reagents (1:10,000). See Supplementary Fig. 8 for uncropped scans of western blots.

**Statistical analysis.** Statistical analysis was performed using unpaired Student's *t*-test. Each experiment was repeated independently at least three times. Comparison of Kaplan–Meier survival plots was performed using Log-rank or Breslow–Gehan–Wilcoxon and for evaluation of statistical difference in frequency distributions the $\chi^2$ (Fisher's exact) analysis algorithm was used.

**Data availability.** The data that support the findings of this study are available on request from the corresponding author (A.V.).

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

## Acknowledgements

We are grateful to K. Rossi, C. Soratroi, I. Gaggl and M. Fischer for excellent technical assistance and animal care. We also thank J. Adams, S. Elledge and T. Mak for mice; S. Tuzlak and L. Fava for helpful discussion. This work was supported by the FWF-funded Doctoral College "Molecular Cell Biology and Oncology" (W1101), grant # P 26856 and grant # I1298, as well as the 'Österreichische Krebshilfe Tirol'. F. Schuler is supported by a Doc-fellowship from the Austrian Academy of Science (ÖAW).

## Author contributions

F.S. performed experiments, analyzed data, contributed to writing and prepared Figures, J.G.W., S.E.L. and S.F.S. performed experiments, P.M., M.L., S.H., V.L. and S.G., generated cell lines, RNA data and/or provided essential reagents and/or edited the manuscript, A.V. designed research, analyzed data, wrote paper, conceived study.

## Additional information

**Competing interests:** The authors declare no competing financial interests.

