## [Peer Review File · Nature Communications]

Reviewers' comments:

Reviewer #1 (Remarks to the Author):

Overall

The investigators have posed interesting questions concerning the role of Chk1 as a tumor suppressor or oncogene. Does it act as both at different times and in different situations? They weigh the information on both sides of the question, i.e., previous results in heterozygous KO mice that might indicate a suppressor role and information on over-expression of Chk1 in many types of tumors that could suggest an oncogenic role, particularly in hematopoietic cancers that express very high levels of Chk1); an example is Burkitt Lymphoma in the Emu-Myc model. They also review the role of Chk1 in cell cycle control and in cell cycle progression after DNA damage. They cite several papers showing that Chk1 inhibitors can be an effective therapeutic agent alone or in combination with other drugs in specific cancers. The question they introduce into the discussion in of all the above is: what is the role of Chk1 in normal B-cell development, and depending on this answer does it suggest complications for Chk1 ablation as a therapeutic agent.

Because of the introduction of so much background information, not in the most logical order, this is a difficult MS to read and to review.

Basically the authors do find that Chk1 is essential for normal mouse B cell development. Several experiments show that there is a block in differentiation or cell loss at the pro to pre-B cell transition. They also investigated the impact of graded inhibitor concentrations on bone marrow derived pre-B cells and mature B-cells from spleen of wt and Vav-BCL2 transgenic mice, with the latter expressing BCL2 throughout the hematopoietic system. IN this system the proliferating pre-B and mature B cells also undergo increased cell death due to inhibition of Chk1. These and other experiments showed that Chk1 is essential fro normal B cell development and allows survival and proliferation of pre-B cells, that encounter a lot of physiological DNA damage during B cell receptor rearrangement and replication stress during rapid expansion.

The conclusion seems to be that Chk1 inhibition can be be a double-edged sword in treatment by Chk1 inhibitors because the very reason its abrogation functions in killing tumor cells is the reason its expression is needed for B cell development as well as for transformation by Myc in B cells, i.e., it is required in cells to deal with replication stress.

Does this mean that Chk1 inhibition treatments will inevitably lead to serious side effects for patients and/or predict ways that the tumor cells may become resistant to the treatment?

The authors propose that there is a window of opportunity for treating Chk1i sensitive tumors without inducing severe side effects associated with complete loss of Chk1 activity and their experiments suggest ways this might be achieved but certainly more experiments in models will be needed. This study lays the groundwork for future experiments.

Conclusions

This report, though describing complex and important pathways and written in what to this reviewer was a rather confusing jumble, does approach an important problem with useful experiments. Perhaps a couple of model drawings to illustrate the pathways described would have helped to make the points much more clearly. Most of all, preclinical treatment models to test the idea that a window of opportunity can be designed to show strong results without serious side effects, would very much strengthen the story, even though this could not possibly cover all possible treatment strategies.

Reviewer #2 (Remarks to the Author):

In this manuscript, Schuler et al evaluate the importance of the Ser/Thr kinase CHK1 in normal B cell development and in B lymphoma cell lines survival. They demonstrate that a pharmacological

inhibitor of CHK1 kills lymphoma cells in an apoptotic dependent way in vitro, and that invalidation of CHK1 in the B lymphocyte lineage compromises both B cell development and myc-driven lymphomagenesis in murine transgenic models. Although there are interesting, original, and potentially important data in this work, in particular genetic evidences identifying CHK1 as an actor of normal B cells development, several issues need to be completed or clarified as stated below.

Figure 1 : the demonstration that CHK1 is essential for lymphoma cells survival is far from being convincing. First the authors used a CHK1 inhibitor at concentrations much more higher than described in the litterature, including in lymphoma cell lines (see for instance Derenzini et al, Oncotarget, 2015). In some studies, maximal effects on CHK1 inhibition are reached with 100 nM of this compound, while the authors used a 1.5 μ M concentration in their experiments in figure 1. Dose-response experiments investigating which concentrations of PF-477736 give rise to CHK1 inhibition in these cell lines are needed. Confirming these data with a second CHK1 inhibitor (there are plenty commercially available, including some currently tested in clinical trials for hematological malignancies) is also necessary. Finally, decreasing CHK1 protein level in these cells through RNA interference experiments is definitively needed in order to conclude that CHK1 is essential for lymphoma cells survival, and that its inhibition kills these cells. In addition, a picture of CHK1 protein levels, CHK1 activation (Ser 345 and/or Ser 317 phosphorylation) and myc expression in the different cell lines would help to better understand the results presented in this figure an in the next ones.

As a minor point, estimation of cell death through sub-G1 population level is not the most adequate method to my point of view. Why did not the authors use annexin V labelling as in Figure 3 ?

Figure 2 : in Figure 2A, could the authors investigate CHK1 protein level (by westen blot or FACS analysis) in the different B cells stages ? There are many post-transcriptional regulations described in the litterature that may affect CHK1protein level, implicating that mRNA expression does not always reflect protein status. Also, I do not see statistical data for this figure, although the authors mentioned n=4 in the figure legend. CHK1 activating phosphorylations (Ser 345 and Ser 317), myc expression, and H2AX phosphorylation could also be investigated in the same fractions.

Figure 3 : these are again in vitro experiments that need to be confirmed by RNA interference-mediated down-regulation of CHK1. Concentrations of the inhibitor used in these experiments are different (and more adequate ?) than in figure 1 ; why did the authors use PF-477736 at 1.5 μ M on lymphoma cell lines (figure 1), and within a 100-500 nM concentration range on primary B cells ?

There is also a problem with the Y axis legend in these figures. When analyzing the data as they are presented, the conclusion is that treatment of B cells with the CHK1 inhibitor decreases apoptotic cell death, which is at the opposite of the authors assertion.

Figure 4 : what is the conclusion of the authors considering these experiments ? As far as I understand, BCL2 overexpression does not apparently restore normal B cells development in CHK1 deficient mice. Did the authors conclude from these data that CHK1 is also involved in B cells proliferation/expansion ? If this is the case, demonstration that CHK1 is important for B cells proliferation is far from being convincing. Could the authors design experiments to directly ask this question ? Which described functions of CHK1 during unperturbed cell cycle (replication, mitosis, ...) could be involved here ? Strengthening this point through experimental data is crucial, since this would represent one of the important and original messages of this work.

To my point of view, the sentence « Taken together... during their rapid extension » (page 7 of the manuscript) is clearly an over-interpretation of the results presented in this paragraph. I do not see any direct experimental evidence that CHK1 is important for B cells proliferation or that replicative stress is somehow involved in this process.

As a general matter, there is a clear discrepancy between results in figure 3 and in figure 4, and the authors explanations for this discrepancy are not sustained by experimental data

Figure 5 : Although the data dealing with myc-dependent lymphomagenesis are interesting, they suffer from the fact that myc is a regulator of CHK1 expression. This somehow interferes with data interpretation, since CHK1 protein level is not only dependent on CHK1 genetic status in the mice and is variable in the different samples. For this reason, western blot analysis of CHK1 protein levels in irradiation experiments presented in figure 5C, would be helpful. In complement, CHK1 phosphorylation status as well as H2AX phosphorylation level could be tested simultaneously.

Figure 6 : in this figure, the data related to cell death are not convincing. The only evidence of increased cell death in the CHK1+/- B cells from bone marrow and spleen is supported by an increased level of cleaved PARP in figure 6A which is neither convincing nor quantified. Why did not the authors use a viability test or an annexin V labelling for these experiments ? DNA damage occurring in S phase is somehow more convincing, although the situation is clearly different in the bone marrow and in the spleen. Could the authors comment on that point ? Finally, I have difficulties to understand why BCL2 status impacts on CHK1-dependent γ H2AX variations in figure 6 C and D . Could the authors comment on that point ?

Figure 7 : there are several points that must be either precised or better demonstrated here in order to sustain the conclusions proposed by the authors. First, all these experiments are based on the use of PF-47736 as a CHK1 inhibitor, generating the same remarks as in figure 1 concerning the specificity of the process described in this set of data. Then, I consider that the authors again over-interpret their results when they claim that impairing apoptosis through BCL2 overexpression or BAX/BAK knock-down shifts the response to CHK1 inhibition from cell death to cell cycle arrest. There is no direct evidence of cell cycle arrest in these experiments, and the growth curve presented for instance in figure 7A argues for cell death still being working in response to CHK1 inhibition. A p53 response associated with a modest p21 increase is not sufficient to conclude to cell cycle arrest. Accumulation in G1 that should occur in that case must be checked, and/or BrDU incorporation experiments could be performed to confirm this hypothesis. G1 and G1/S CDK activities as well as pRb phosphorylation status could also be tested.

Reviewer #3 (Remarks to the Author):

Schuler et al tackle an important topic, namely assessing the potential of the Chk1 protein kinase as an anti-cancer target in hematological malignancies and the possible complications that might arise from systemic inhibition of Chk1. Using a combination of genetic and chemical approaches to inactivate Chk1 partially or completely in cell lines and mice they report that: 1) Chk1 inhibition is highly toxic in leukemia cells with high levels of replicative stress and that cell death occurs via a Bak-Bax-dependent mitochondrial apoptosis mechanism that can be inhibited by Bcl-2 overexpression, 2) ablation of Chk1 in the B-cell lineage in vivo blocks B-cell development at the pro-B stage, 3) This block likely results from apoptotic cell death but it cannot be rescued by Bcl-2 overexpression owing to a DNA damage response whose final outcome, cell cycle arrest or polyploidization, depends on p53 status, 4) Chk1 haplo-insufficiency hinders the development of hematological malignancies in vivo in contrast to what has been observed in other murine tissues such as intestine and skin.

Based on these observations the authors propose that hematological malignancies are good candidates for treatment with Chk1 inhibitor drugs, since they seem to both overexpress and depend on Chk1 for proliferation/ survival, but that systemic treatment could risk immunological side-effects or even the possibility of tumor promotion by promoting aneuploidy.

In general the experiments have been carefully designed, executed, and interpreted. The manuscript is very clearly written and the conclusions are well supported by the data. Several novel findings are reported that add to the body of knowledge on this important topic and will be of considerable interest to researchers in the cancer/ genome stability field.

Point to point reply: NCOMMS-17-06928**Reviewer #1:**

Overall: The investigators have posed interesting questions concerning the role of Chk1 as a tumor suppressor or oncogene. Does it act as both at different times and in different situations? They weigh the information on both sides of the question, i.e., previous results in heterozygous KO mice that might indicate a suppressor role and information on over-expression of Chk1 in many types of tumors that could suggest an oncogenic role, particularly in hematopoietic cancers that express very high levels of Chk1); an example is Burkitt Lymphoma in the Emu-Myc model. They also review the role of Chk1 in cell cycle control and in cell cycle progression after DNA damage. They cite several papers showing that Chk1 inhibitors can be an effective therapeutic agent alone or in combination with other drugs in specific cancers. The question they introduce into the discussion in of all the above is: what is the role of Chk1 in normal B-cell development, and depending on this answer does it suggest complications for Chk1 ablation as a therapeutic agent. Because of the introduction of so much background information, not in the most logical order, this is a difficult MS to read and to review. Basically the authors do find that Chk1 is essential for normal mouse B cell development. Several experiments show that there is a block in differentiation or cell loss at the pro to pre-B cell transition. They also investigated the impact of graded inhibitor concentrations on bone marrow derived pre-B cells and mature B-cells from spleen of wt and Vav-BCL2 transgenic mice, with the latter expressing BCL2 throughout the hematopoietic system. IN this system the proliferating pre-B and mature B cells also undergo increased cell death due to inhibition of Chk1. These and other experiments showed that Chk1 is essential for normal B cell development and allows survival and proliferation of pre-B cells, that encounter a lot of physiological DNA damage during B cell receptor rearrangement and replication stress during rapid expansion.

The conclusion seems to be that Chk1 inhibition can be a double-edged sword in treatment by Chk1 inhibitors because the very reason its abrogation functions in killing tumor cells is the reason its expression is needed for B cell development as well as for transformation by Myc in B cells, i.e., it is required in cells to deal with replication stress.

Does this mean that Chk1 inhibition treatments will inevitably lead to serious side effects for patients and/or predict ways that the tumor cells may become resistant to the treatment?

The authors propose that there is a window of opportunity for treating Chk1i sensitive tumors without inducing severe side effects associated with complete loss of Chk1 activity and their experiments suggest ways this might be achieved but certainly more experiments in models will be needed. This study lays the groundwork for future experiments.

Conclusions: This report, though describing complex and important pathways and written in what to this reviewer was a rather confusing jumble, does approach an important problem with useful experiments. Perhaps a couple of model drawings to illustrate the pathways described would have helped to make the points much more clearly. Most of all, preclinical treatment models to test the idea that a window of opportunity can be designed to show strong results without serious side effects, would very much strengthen the story, even though this could not possibly cover all possible treatment strategies.

Response: We thank this referee for the thorough assessment of our work.

Admittedly, this field is very hard to cover in a comprehensive and satisfying manner in a short introduction, given the vast body of data available in the literature. We were hoping that we were able to give a decent overview of current knowledge and provide a rationale for our research questions asked. It was not our primary intention to test the efficacy of CHK1 inhibitors in mouse models of MYC driven lymphoma, as this has been already covered in other studies cited (Ref. 31 & 33) but to understand the role of CHK1 in normal B cell development and lymphocyte transformation in order to get better insight into its physiological role to better exploit its potential as a target in anti-cancer therapy, also by estimating potential side effects.

We hope that the cartoon provided in the revised version of our manuscript (Fig. 8) will help to summarize our findings in a light and digestible manner, separating effects seen in MYC driven lymphoma in vivo from those noted in human cancer cell lines. We hope that this will be appreciated by this referee as it also includes aspects of our discussion how CHK1i treatment can select for resistant and more aggressive clones when cancer cells become apoptosis-resistant.

Reviewer # 2

Comment: In this manuscript, Schuler et al evaluate the importance of the Ser/Thr kinase CHK1 in normal B cell development and in B lymphoma cell lines survival. They demonstrate that a pharmacological inhibitor of CHK1 kills lymphoma cells in an apoptotic dependent way in vitro, and that invalidation of CHK1 in the B lymphocyte lineage compromises both B cell development and myc-driven lymphomagenesis in murine transgenic models. Although there are interesting, original, and potentially important data in this work, in particular genetic evidences identifying CHK1 as an actor of normal B cells development, several issues need to be completed or clarified as stated below.

Response: We are thankful for this referee's overall positive evaluation and the helpful suggestions that have helped to improve the overall quality of our work.

Comment on Figure 1: The demonstration that CHK1 is essential for lymphoma cells survival is far from being convincing. First the authors used a CHK1 inhibitor at concentrations much higher than described in the literature, including in lymphoma cell lines (see for instance Derenzini et al, Oncotarget, 2015). In some studies, maximal effects on CHK1 inhibition are reached with 100 nM of this compound, while the authors used a 1.5 μ M concentration in their experiments in figure 1.

Dose-response experiments investigating which concentrations of PF-477736 give rise to CHK1 inhibition in these cell lines are needed. Confirming these data with a second CHK1 inhibitor (there are plenty commercially available, including some currently tested in clinical trials for hematological malignancies) is also necessary. Finally, decreasing CHK1 protein level in these cells through RNA interference experiments is definitively needed in order to conclude that CHK1 is essential for lymphoma cell survival, and that its inhibition kills these cells. In addition, a picture of CHK1 protein levels, CHK1 activation (Ser 345 and/or Ser 317 phosphorylation) and myc expression in the different cell lines would help to better understand the results presented in this figure and in the next ones.

As a minor point, estimation of cell death through sub-G1 population level is not the

most adequate method to my point of view. Why did not the authors use annexin V labelling as in Figure 3?

Response: We appreciate this concern relating to potential off-target effects of high inhibitor concentrations. We chose sub-G1 analysis over Annexin V binding assays in order to obtain information on the cell cycle distribution of these cells, next to cell death rates, that proved also informative regarding polyploidization of inhibitor treated cells.

To consolidate our finding we have now performed **(i)** titration experiments using the CHK1 inhibitor PF-477736 and assessed survival by Annexin-V-staining and flow cytometry to allow faster and alternative evaluation of cell death rates. A clear dose-dependence can be seen across cell lines (**new Fig. 1B**). Moreover, **(ii)** all these experiments have been repeated using a second inhibitor targeting CHK1, CHIR-124, that appeared more potent at lower concentrations, to confirm our findings (**new Fig. S1**). As suggested, we **(iii)** also performed western analysis to document expression of CHK1 and its activity, using CHK1-specific and CHK1ser345 specific antibodies. Expression of MYC was assessed in parallel (**Fig. 1A**). While all cells do express CHK1, activity varies widely across cell lines. Interestingly, an indirect correlation between CHK1Ser345 levels and cell death rates was noted. Cells with high Ser345 levels required higher inhibitor levels to be killed. Our findings are presented in the according results section. **(iv)** To confirm these results by genetic means, we have generated BL-2 Burkitt and Nalm-6 ALL cell lines as representative examples for both tumor types that harbor a conditional shRNA targeting CHK1. Depletion of CHK1 by doxycycline addition coincided with increased PARP1 cleavage, a marker for Caspase-3 activation, indicating the induction of cell death. Results are displayed in the new version of Fig. S1.

Comment on Figure 2: in Figure 2A, could the authors investigate CHK1 protein level (by western blot or FACS analysis) in the different B cells stages? There are many post-transcriptional regulations described in the literature that may affect CHK1 protein level, implicating that mRNA expression does not always reflect protein status. Also, I do not see statistical data for this figure, although the authors mentioned n=4 in the figure legend. CHK1 activating phosphorylations (Ser 345 and Ser 317), myc expression, and H2AX phosphorylation could also be investigated in the same fractions.

Response: We were unable to provide error bars in Fig. 2A, as these values were extracted from deep sequencing data that has been performed using pooled RNA samples from four individual animals (Ref # 37). However, we confirmed these data by **(i)** performing qPCR analyses on mRNA from freshly sorted B cell populations, identical to those used in the NGS analysis. Moreover, we have **(ii)** performed western analysis on these cells. These experiments corroborate our initial findings showing that CHK1 mRNA is a good read out for protein levels at the different developmental stages explored. The S345 antibody, however, was not suited to read out activation status in mouse cell extracts. In conclusion, CHK1 expression correlates perfectly well with proliferation state of these cells, that is high in early pro/pre B cells (pooled for western, due to low numbers available from cell sorting) and in CD40-activated mature B cells, but low in immature or resting mature B cells (**Fig. 2A,B**). MYC levels were assessed by qPCR but did not directly correlate with

CHK1 mRNA levels (not shown). Protein levels were evaluated by western, but ultimately not detectable with the antibodies available to us, as these proved to be human specific.

Comment on Figure 3: these are again in vitro experiments that need to be confirmed by RNA interference-mediated down-regulation of CHK1. Concentrations of the inhibitor used in these experiments are different (and more adequate?) than in figure 1 ; why did the authors use PF-477736 at 1.5 μ M on lymphoma cell lines (figure 1), and within a 100-500 nM concentration range on primary B cells ?

There is also a problem with the Y axis legend in these figures. When analyzing the data as they are presented, the conclusion is that treatment of B cells with the CHK1 inhibitor decreases apoptotic cell death, which is at the opposite of the authors assertion.

Response: To avoid confusion in data presentation, we have now changed the axis label to % survival which was assessed by Annexin V / PI staining and flow cytometry.

To explain the use of the different inhibitor concentrations, we can say that we have pre-titrated CHK1i on primary B cells beforehand and noted that they were more sensitive than most lymphoma lines tested, hence, we started experiments with lower inhibitor doses.

Nonetheless, to corroborate our finding we have now also tested CHIR-124 for its potential to target primary B cells. This inhibitor, again, proved more effective when compared to PF-477736 (**Fig. S4**). In summary, both inhibitors potently kill immature (cycling-competent) pre B cells in a BCL2-dependent manner (**Fig. 3, S4**), while PF-477736 is also killing resting splenic B cells at 500nM, suggesting potential off-target effects might account for their BCL2 regulated cell death at high inhibitor concentrations. Of note, 500nM CHIR-124 also killed BCL2-transgenic B cells, indicating that this death was no longer apoptotic (not shown) and we decided not to display this result to avoid confusion. Yet, together, we believe that these experiments document a CHK1-dependence of mouse primary B cell progenitors for survival and subsequent differentiation, while resting B cells, as expected are largely independent of CHK1 function for survival.

Comment on Figure 4: what is the conclusion of the authors considering these experiments? As far as I understand, BCL2 overexpression does not apparently restore normal B cells development in CHK1 deficient mice. Did the authors conclude from these data that CHK1 is also involved in B cells proliferation/expansion? If this is the case, demonstration that CHK1 is important for B cells proliferation is far from being convincing. Could the authors design experiments to directly ask this question?

Which described functions of CHK1 during unperturbed cell cycle (replication, mitosis, ...) could be involved here? Strengthening this point through experimental data is crucial, since this would represent one of the important and original messages of this work. To my point of view, the sentence « Taken together... during their rapid extension » (page 7 of the manuscript) is clearly an over-interpretation of the results presented in this paragraph. I do not see any direct experimental evidence that CHK1 is important for B cells proliferation or that replicative stress is somehow involved in this process.

As a general matter, there is a clear discrepancy between results in figure 3 and in figure 4, and the authors explanations for this discrepancy are not sustained by experimental data

Response: These comments listed above are all justified. There is a clear discrepancy between the data in Figure 3, showing that CHK1i kills B cells in a BCL2 dependent manner in culture and Figure 4 where we document that overexpression of BCL2 fails to restore B cell development in mice were we have deleted CHK1 using CRE recombinase at the pre-B cell stage. We try to line out our rational better below.

We conclude from these experiments, and data in Fig. 7, that CHK1 is essential for the normal proliferation and survival of these cells. We believe our data supports a model shown now in **Fig. 8** that there is an interdependence of these two processes. This is supported by the notion that cells that cannot die in response to CHK1 inhibition because they overexpress exogenous BCL2 or lack BAX/BAK, do trigger a p53 response and upregulate p21 in an attempt to halt cell cycle progression (**Fig. 7**). This block can be leaky in cancer cells, in particular those lacking p53, but appears robust in mouse B cells (**Fig. S1, Fig. 7**).

In further support of this hypothesis we now also present data that shows the dependence of pro-B cells on CHK1 for proliferation **and** survival. To this end we have sorted pro-B cells from *Chk1*^{+/-} animals and expanded them in the presence of IL7 which also allows progression from the pro- to the pre-B cell stage. Loss of one allele of *Chk1* clearly limits the proliferative capacity of these cells, generating fewer pre-B cells in the presence of IL7 and renders these pre-B cells also more susceptible to CHK1 inhibitor (**Fig. 3A**).

Overall, we believe that these experiments provide strong evidence that there is a clear interdependence and that CHK1 is essential for normal proliferation of early B cells and B cell lymphomas and their survival. With the tools at hand, we were not able to unambiguously dissect if the S-phase or G2/M-phase function of CHK1 is key in either biological response (proliferation vs. survival), but ultimately, it seems clear that reduced expression of CHK1 (e.g. in *Chk1*^{+/-} cells ± MYC) or chemical inhibition (CHK1i) causes increased levels of DNA damage in cycling cells, as indicated by g-H2AX staining (**e.g. Fig. 6B,C; Fig. 7**) that culminates in increased cell death susceptibility (**Fig. 3A & 6A**). When cell death is blocked, this gradually deviates the response to cell cycle arrest, which involves p53 activation in response to DNA damage. Proficiency of this arrest depends on the cancer cell line tested (**Fig. 7**). When p53 is inactive, e.g., in Burkitt lymphoma cells, these cells do not divide but do show a trend to become polyploid in response to CHK1i, most likely due to cytokinesis failure that can be an indirect consequence of CHK1 inhibition, leading to impaired AuroraB function (**Fig 7C and Fig. S1**), as reported by others (Ref. 44). This poses a potential threat to patients showing p53 mutations and will be treated with such inhibitors, as this can foster aneuploidy and select for complex and potentially more aggressive karyotypes.

Comment on Figure 5: Although the data dealing with myc-dependent lymphomagenesis are interesting, they suffer from the fact that myc is a regulator of CHK1 expression. This somehow interferes with data interpretation, since CHK1 protein level is not only dependent on CHK1 genetic status in the mice and is variable in the different samples. For this reason, western blot analysis of CHK1

protein levels in irradiation experiments presented in figure 5C, would be helpful. In complement, CHK1 phosphorylation status as well as H2AX phosphorylation level could be tested simultaneously.

Response: We appreciate this concern, yet, our long-lasting experience working with gene-modified mice has told us that there is a substantial variation in expression levels of any given target gene we investigated so far. This depends on many variables, including gender, age, housing conditions. What remains important here is that even in the presence of MYC, that increases levels of CHK1 dramatically (Fig. 5A), B cells from MYC/CHK1^{+/-} mice, on average, do have lower levels than B cells from MYC/CHK1^{+/+} mice, the two genotypes relevant in our tumor cohort. Hence, it is fair to believe that the delay seen in cancer onset is due to differences in Chk1 protein levels. To document the biological variation in the system more clearly, we now show more examples in the new version of **Fig. 5A**.

Moreover, we provide now evidence that CHK1 levels are increased in hematopoietic organs from mice exposed to low dose radiation during their recovery phase, as assessed on day 4 after radiation, supporting our overall hypothesis that replication stress-management by relies on proficient levels of CHK1 in order to prevent disease onset, also in thymic lymphomas, induced by irradiation damage (**Fig. 5C**). As mentioned above the Ser345 antibody did not work well in our hands when using mouse material.

Comment on Figure 6: in this figure, the data related to cell death are not convincing. The only evidence of increased cell death in the CHK1^{+/-} B cells from bone marrow and spleen is supported by an increased level of cleaved PARP in figure 6A which is neither convincing nor quantified. Why did not the authors use a viability test or an annexin V labelling for these experiments?

DNA damage occurring in S phase is somehow more convincing, although the situation is clearly different in the bone marrow and in the spleen. Could the authors comment on that point?

Response: We can relate to these comments very well. We have made several efforts to better document increased cell death rates in MYC overexpressing B cells lacking one allele of CHK1. As can be seen in **Fig. S6** of our original submission, FACS-sorted B cells overexpressing MYC do die rapidly in culture and we were unable to document further cell death acceleration due to loss of one allele of *Chk1*. Therefore, we were relying on biochemical characterization of cell death markers in cells lysed immediately after sacrifice to find support for our hypothesis, shown in **Fig. 6A**. The difference between BM and spleen was most likely due to the fact that we have sorted cells from BM, to enrich for pre-B cells, while we lysed total spleen. This may have blurred the effects in immature splenic B cells that are the target of MYC driven transformation. Despite some attempts, we were unable to improve the quality of the data generated from spleen. Overall, again, we see the same trend, i.e., an increase in PARP1 cleavage even though the effects are simply not as clean as in pre-B cells. We discuss the limitations our findings in more detail on page 9.

Finally, I have difficulties to understand why BCL2 status impacts on CHK1-dependent γ H2AX variations in figure 6 C and D. Could the authors comment on that point?

Response: There must be a misunderstanding, as in panel 6C and 6D only the loss of one allele of Chk1 or the addition of a CHK1 inhibitor leads to increased DNA damage, as read out by gH2AX staining by FACS. BCL2 overexpression is common to **all** these cells to exclude confounding effects due to apoptosis induction upon inhibitor treatment that will inevitably cause gH2AX reactivity downstream of caspase-mediated activation of DNaseI.

Figure 7: there are several points that must be either precised or better demonstrated here in order to sustain the conclusions proposed by the authors. First, all these experiments are based on the use of PF-477736 as a CHK1 inhibitor, generating the same remarks as in figure 1 concerning the specificity of the process described in this set of data.

Response: We have included additional experiments, assessing proliferation and population doublings using both CHK1 inhibitors at different doses, revealing comparable results as reported initially and in support of our overall conclusion (**Fig. 7**).

Then, I consider that the authors again over-interpret their results when they claim that impairing apoptosis through BCL2 overexpression or BAX/BAK knock-down shifts the response to CHK1 inhibition from cell death to cell cycle arrest. There is no direct evidence of cell cycle arrest in these experiments, and the growth curve presented for instance in figure 7A argues for cell death still being working in response to CHK1 inhibition. A p53 response associated with a modest p21 increase is not sufficient to conclude to cell cycle arrest. Accumulation in G1 that should occur in that case must be checked, and/or BrDU incorporation experiments could be performed to confirm this hypothesis. G1 and G1/S CDK activities as well as pRb phosphorylation status could also be tested.

Response: We appreciate these comments and feel that our way of presenting and interpreting our data may have been inconsistent and somewhat overambitious. Clearly, it cannot be excluded that high concentrations of CHK1i can also kill BCL2 overexpressing cells. In fact, mouse-derived pre-B cells overexpressing MYC+BCL2 (**Fig. 7A**), do seem to eventually die in response to 200nM of CHIR-124, while at 100nM CHIR-124 or in response to PF-477736 these cells stop to expand. On the other hand, BAX/BAK DKO stop to divide only at high doses of inhibitor (**Fig. 7B**) and become polyploid (**Fig. S1F**), similar to what is seen in BCL2 overexpressing Burkitt lymphoma lines (**Fig. 7C, Fig. S1F**).

We also agree that p53 and p21 induction per se does not always suffice to promote cell cycle arrest. We actually do see an accumulation of BL-2 cells in G2/M, rather than G1 which we interpret as cells that arrest as tetraploid G1 cells that failed cytokinesis. Considering the polyploidy see in other p53 deficient Burkitt lymphoma cells as well as BAX/BAK DKO cells that continue to endoreduplicate their DNA we conclude that these cells are at risk to develop (higher levels of) aneuploidy which certainly has implication for treatment efficacy. We discuss this issue in more detail on page 9 and 13.

Reviewer #3

Schuler et al tackle an important topic, namely assessing the potential of the Chk1 protein kinase as an anti-cancer target in hematological malignancies and the possible complications that might arise from systemic inhibition of Chk1. Using a combination of genetic and chemical approaches to inactivate Chk1 partially or completely in cell lines and mice they report that: 1) Chk1 inhibition is highly toxic in leukemia cells with high levels of replicative stress and that cell death occurs via a Bak-Bax-dependent mitochondrial apoptosis mechanism that can be inhibited by Bcl-2 overexpression, 2) ablation of Chk1 in the B-cell lineage in vivo blocks B-cell development at the pro-B stage, 3) This block likely results from apoptotic cell death but it cannot be rescued by Bcl-2 overexpression owing to a DNA damage response whose final outcome, cell cycle arrest or polyploidization, depends on p53 status, 4) Chk1 haplo-insufficiency hinders the development of hematological malignancies in vivo in contrast to what has been observed in other murine tissues such as intestine and skin.

Based on these observations the authors propose that hematological malignancies are good candidates for treatment with Chk1 inhibitor drugs, since they seem to both overexpress and depend on Chk1 for proliferation/ survival, but that systemic treatment could risk immunological side-effects or even the possibility of tumor promotion by promoting aneuploidy.

In general, the experiments have been carefully designed, executed, and interpreted. The manuscript is very clearly written and the conclusions are well supported by the data. Several novel findings are reported that add to the body of knowledge on this important topic and will be of considerable interest to researchers in the cancer/ genome stability field.

Response: We would like to thank this referee for his/her careful and overall very encouraging comments on our work. As this referee did not suggest any additional experiments, we hope that the revisions made based on input by Referee#2 will be appreciated by this reviewer.

REVIEWERS' COMMENTS:

Reviewer #2 (Remarks to the Author):

In the revised version of their manuscript entitled « Checkpoint kinase 1 (CHK1) controls normal B cell development, lymphomagenesis and cancer cell survival », Schuler et al have considerably improved the quality of their demonstration that CHK1 is a key determinant of Pre-B cells proliferation and differentiation and of lymphomagenesis. However, there are still a few concerns that need to be resolved before this can be accepted for publication.

One point deals with the supposed functional link between myc and CHK1 expression, which to my point of view remains poorly convincing when considering the corresponding data in the manuscript. There is no apparent correlation between myc and CHK1 protein levels in figure 1A, and H2AX phosphorylation also does not seem to be related with myc level, although this factor is supposed to induce replicative stress. In figure 2A, the right panel (Q-PCR analysis) does not convincingly fit with the data presented in the left panel (RNA seq), and I did not find any legend for the black and white bars in this figure (CHK1 and myc respectively ?). As a general matter, the importance of myc for Chk1 regulation both in lymphoma cells and during B cells differentiation must be precised and adjusted all along the manuscript, since the data presented in figures 1 and 2 do not argue for this relationship.

The western blot presented in new figure 2B convincingly demonstrates that CHK1 protein is highly expressed in Pre/pro-B cells and is dramatically reduced in more differentiated stages. However the amplitude of this variation at the protein level is much more important than for the mRNA (which is only reduced by a half as shown in immature B cells in figure 2A). This discrepancy probably reflects a much more complex regulation of CHK1 than simple transcriptional regulation by myc, probably involving different types of post-transcriptional events. A discussion of this point in the « Results » and « Discussion » sections must be included, since the authors claim that CHK1 mRNA and protein variations are well correlated in the present version of the manuscript.

Finally, in their answering letter, the authors claim that the sensitivity of lymphoma cell lines to CHK1 inhibition correlate with the activation (Ser 345 phosphorylation) status of CHK1, but I do not observe such a correlation in the figure. For instance the most sensitive cell line BL41 presents a low level of CHK1 phosphorylation, while the less sensitive one (Raji) has a high level of Ser 354 phosphorylation. I did not find the same remark in the text of the paper, but the authors must check that they did not mention that point somewhere in their manuscript.

Once these different points taken into consideration, the manuscript will be suitable for publication in Nature Communications.

Point-To-Point Reply:

Comment: In the revised version of their manuscript entitled « Checkpoint kinase 1 (CHK1) controls normal B cell development, lymphomagenesis and cancer cell survival », Schuler et al have considerably improved the quality of their demonstration that CHK1 is a key determinant of Pre-B cells proliferation and differentiation and of lymphomagenesis. However, there are still a few concerns that need to be resolved before this can be accepted for publication. One point deals with the supposed functional link between myc and CHK1 expression, which to my point of view remains poorly convincing when considering the corresponding data in the manuscript. There is no apparent correlation between myc and CHK1 protein levels in figure 1A, and H2AX phosphorylation also does not seem to be related with myc level, although this factor is supposed to induce replicative stress.

In figure 2A, the right panel (Q-PCR analysis) does not convincingly fit with the data presented in the left panel (RNA seq), and I did not find any legend for the black and white bars in this figure (CHK1 and myc respectively ?). As a general matter, the importance of myc for Chk1 regulation both in lymphoma cells and during B cells differentiation must be precised and adjusted all along the manuscript, since the data presented in figures 1 and 2 do not argue for this relationship.

Reply: We acknowledge the viewpoint of this reviewer and have amended the text accordingly on page 6 by amending the text to: *While these analyses showed a clear correlation of CHK1 protein levels with proliferation status, Chk1 mRNA expression correlated only poorly with levels of c-Myc across B cell development (Fig. 2A), contrasting a previously reported interrelationship between these two genes*^{31, 38}.

We are uncertain about the comment on gammaH2AX, as we never compared MYC positive vs. MYC negative cells for that marker of DNA damage, hence, never claimed a correlation between both parameters, yet demonstrate that a reduction of CHK1, either genetically (*Chk1*^{+/-} cells) or chemically (CHK1i treatment), leads to increased gH2AX levels, in support of a potential link between loss of CHK1 activity and increased DNA damage in these cells.

The western blot presented in new figure 2B convincingly demonstrates that CHK1 protein is highly expressed in Pre/pro-B cells and is dramatically reduced in more differentiated stages. However, the amplitude of this variation at the protein level is much more important than for the mRNA (which is only reduced by a half as shown in immature B cells in figure2A). This discrepancy probably reflects a much more complex regulation of CHK1 than simple transcriptional regulation by myc, probably involving different types of post-transcriptional events. A discussion of this point in the « Results » and « Discussion » sections must be included, since the authors claim that CHK1 mRNA and protein variations are well correlated in the present version of the manuscript.

Reply: We no longer make this claim, as indicated above. Hence, we believe such a discussion is no longer needed.

Comment: Finally, in their answering letter, the authors claim that the sensitivity of lymphoma cell lines to CHK1 inhibition correlate with the activation (Ser 345

phosphorylation) status of CHK1, but I do not observe such a correlation in the figure. For instance, the most sensitive cell line BL41 presents a low level of CHK1 phosphorylation, while the less sensitive one (Raji) has a high level of Ser 354 phosphorylation. I did not find the same remark in the text of the paper, but the authors must check that they did not mention that point somewhere in their manuscript.

Reply: This referee is correct in his claim. If anything there is a reverse correlation – high Ser345 phosphorylation correlates with reduced sensitivity to inhibitor. Given the low number of cell lines tested we refrain from drawing any conclusion. We have one more time checked the manuscript and did not find any related claim, as acknowledged by this reviewer already.

Once these different points taken into consideration, the manuscript will be suitable for publication in Nature Communications.

Reply: We are pleased to hear our revised version convinced also this referee and we want to thank him/her for the time taken to help improve our work.